# Reduced biomass burning emissions reconcile conflicting estimates of the post-2006 atmospheric methane budget

John R. Worden [1], A. Anthony Bloom[1], Sudhanshu Pandey[2,3], Zhe Jiang[1,4], Helen M. Worden[4], Thomas W. Walker[1], Sander Houweling[2,3,5] & Thomas Röckmann[2]

Several viable but conflicting explanations have been proposed to explain the recent ~8 p.p.b. per year increase in atmospheric methane after 2006, equivalent to net emissions increase of ~25 Tg $CH_4$ per year. A concurrent increase in atmospheric ethane implicates a fossil source; a concurrent decrease in the heavy isotope content of methane points toward a biogenic source, while other studies propose a decrease in the chemical sink (OH). Here we show that biomass burning emissions of methane decreased by 3.7 (±1.4) Tg $CH_4$ per year from the 2001–2007 to the 2008–2014 time periods using satellite measurements of CO and $CH_4$, nearly twice the decrease expected from prior estimates. After updating both the total and isotopic budgets for atmospheric methane with these revised biomass burning emissions (and assuming no change to the chemical sink), we find that fossil fuels contribute between 12–19 Tg $CH_4$ per year to the recent atmospheric methane increase, thus reconciling the isotopic- and ethane-based results.

---

[1] Jet Propulsion Laboratory, California Institute for Technology, Pasadena, 91109 CA, USA. [2] Institute for Marine and Atmospheric Research Utrecht, Utrecht University, Utrecht, The Netherlands. [3] SRON Netherlands Institute for Space Research, Utrecht, The Netherlands. [4] National Center for Atmospheric Research, Boulder, 80301 CO, USA. [5] Department of Earth Sciences, Vrije Universiteit Amsterdam, Amsterdam, The Netherlands. Correspondence and requests for materials should be addressed to J.R.W. (email: john.r.worden@jpl.nasa.gov)

Recent changes in the growth rate of methane[1], the second most important greenhouse gas, and important ozone precursor[2], could be due to changing anthropogenic emissions in the form of fossil fuel (FF) or agricultural emissions[3–8]. Alternatively, natural wetland methane fluxes in the high latitudes or tropics could be increasing in response to variations in temperature, the water cycle, and/or carbon availability to methanogens[9–12], giving a preview of carbon cycle feedbacks to global warming[13]. However, determining the relative contributions of anthropogenic, biogeochemical, and chemical drivers of methane trends has been extremely challenging and consequently there is effectively no confidence in projections of future atmospheric methane concentrations. The striking disagreement from several recent studies explaining the changes to atmospheric methane since 2006[5–8] is likely due to the assumptions (and extrapolations) involved in attributing source variability to the observed changes in atmospheric methane. For example, surface measurements of $CH_4$ and its isotopic composition suggest a shift of methane sources toward increasing tropical biogenic (BG) sources[5,14,15]. However, this explanation appears to directly conflict with observations of increasing FF sources that range between 5 and 25 Tg $CH_4$ per year based on ethane/$CH_4$ ratios[6–8] as well as studies based on satellite-based total column methane measurements[16,17]. Other studies[18,19] show that we cannot rule out inter-annual variations in the hydroxyl radical (OH) chemical methane sink as the cause; however, these studies do not directly show changes in atmospheric OH or provide a mechanistic reason for a change.

Biomass burning (BB) contributes only moderately to atmospheric methane with past estimates ranging from 14 to 26 Tg $CH_4$ per year out of the ~550 Tg $CH_4$ per year budget[20,21]. The range of BB $CH_4$ emissions estimates is in part due to uncertainties in burnt area estimates, combustion factors, and emission factors[22–25] and to large inter-annual variability (IAV) resulting from substantial regional changes in rainfall due to ENSO[26]. For example, larger than normal inter-annual changes in atmospheric $CH_4$ in 2006 and likely 1997 can be directly attributed to massive Indonesian peat fires[27,28]. Estimates based on burnt area suggest a decrease of ~2 Tg per year after 2007 (Global Fire Emissions Database, version 4—GFEDv4s)[29] with decreasing burnt area over Africa likely due to better fire management and agricultural practices[30] as well as reduced emissions over South America and Indonesia[25,31,32]. Our study focuses on how changes in biomass burning BB emissions of methane affect our knowledge of the FF and BG components of the atmospheric methane budget.

GFED bottom-up estimates for methane emissions from BB depend on satellite observations of burnt area, vegetation type, combustion efficiency, and amount of burnt biomass[29,33]. Top-down estimates depend on the combination of observationally constrained total CO flux estimates and in situ or satellite constraints on the $CH_4$/CO ratio[25,28] (Methods). Because the seasonality and location of fires are typically distinct from other emissions such as biofuels, industry, and transportation, top-down approaches can robustly distinguish biomass burning emissions from other sources based on satellite CO concentration measurements and prior information on burnt-area-based fire emissions estimates[25,28,31,32]. Here, we combine bottom-up estimates of fire emissions, based on burnt area measurements, with the top-down CO emissions estimates[31] (Methods), based on the satellite concentration data and the adjoint of the Goddard Earth Observing System Chemistry model (GEOS-Chem). This approach for quantifying CO and $CH_4$ fire emissions accounts for published uncertainties in the bottom-up estimates and includes empirical estimates of the key factors that contribute to uncertainties in emissions inferred from concentration data such as errors in transport and chemistry, partitioning of CO emissions

on the $5 \times 4°$ GEOS-Chem grid cell to FF, fires, or chemical sources[31,32], and uncertainties in the $CH_4$/CO emission factors and their IAV. We use satellite and in situ measurements of $CH_4$/CO ratios to evaluate fire-based $CH_4$/CO values and their associated uncertainties (Methods). We then show that biomass burning emissions of methane decreased by 3.7 ($\pm1.4$) Tg $CH_4$ per year from the 2001–2007 to the 2008–2014 time periods, nearly twice the decrease expected from prior estimates based on burnt area measurements. After updating both the total and isotopic budgets for atmospheric methane with these revised biomass burning emissions (and assuming no change to the chemical sink), we find that FFs and BG sources contribute 12–19 Tg $CH_4$ per year and 12–16 Tg $CH_4$ per year, respectively, to the recent atmospheric methane increase, thus reconciling the isotopic- and ethane-based results.

## Results

**Trend in $CH_4$ emissions from fires.** Figure 1 shows the time series of $CH_4$ emissions that were obtained from GFEDv4s and top-down estimates based on CO emission estimates and GFED4s-based emission ratios. The CO-based fire $CH_4$ emissions estimates amount to $14.8 \pm 3.8$ Tg $CH_4$ per year for the 2001–2007 time period and $11.1 \pm 3$ Tg $CH_4$ per year for the 2008–2014 time period, with a $3.7 \pm 1.4$ Tg $CH_4$ per year decrease between the two time periods. The mean burnt area (a priori)-based estimate from GFED4s is slightly larger and shows a slightly smaller decrease (2.3 Tg $CH_4$ per year) in fire emissions after 2007 relative to the 2001–2006 time period. The range of uncertainties (shown as blue error bars in Fig. 1 is determined by the uncertainty in top-down CO emission estimates that are derived empirically using the approaches discussed in the Methods). The red shading describes the range of uncertainty stemming from uncertainties in $CH_4$/CO emission factors (Methods). By assuming temporally constant sector-specific $CH_4$/CO emission factors, we find that mean 2001–2014 emissions average to $12.9 \pm 3.3$ Tg $CH_4$ per year, and the decrease averages to $3.7 \pm 1.4$ Tg $CH_4$ per year for 2008–2014, relative to 2001–2007. This decrease is largely accounted for by a $2.9 \pm 1.2$ Tg $CH_4$ per year decrease during 2006–2008, which is primarily attributable to a biomass burning decrease in Indonesia and South America[25,28,31].

While we account for the IAV in the global $CH_4$/CO emission factors due to varying contributions from individual fire types (such as savannas or peat fires), the temporal $CH_4$/CO variability due to underlying combustion processes for each fire type is currently not well characterized. We assess the sensitivity of our result on decreasing methane BB emissions to larger IAV in global $CH_4$/CO emission factors by randomly perturbing annual sector-specific $CH_4$/CO emission factors (Methods) and examining how they affect 2001–2014 BB methane emission trends. We find that the probability of a decrease in methane BB emissions throughout 2001–2014 is >95% assuming that any unexplained global annual $CH_4$/CO variability is <21% (Fig. 2). There is a 95% probability that fire methane emissions during 2008–2014 decreased relative to 2001–2007 if the IAV of the global annual $CH_4$/CO ratio is <32%. These perturbations to the $CH_4$/CO emission factors are roughly a factor of three greater than expected variability from changes in fire-type contributions alone (global $CH_4$/CO IAV 7–8%, Fig. 2). We therefore conclude that the decrease in biomass burning emissions of methane after 2007 cannot be easily explained by unaccounted inter-annual variations of the $CH_4$/CO due to errors in fire-type contributions. Furthermore, since coherent sector-specific $CH_4$/CO inter-annual variations comparable to within-sector $CH_4$/CO uncertainty (gray area in Fig. 2) are improbable, unaccounted inter-annual sector-

specific $CH_4/CO$ variations cannot easily explain the biomass burning emission trends.

Wetter years associated with La Nina during the 2008 through 2014 time periods likely contributed to the observed decrease in fire emissions in South America and Indonesia[25,31]. It is also likely that this increased precipitation in these regions affects the fuel moisture content and in turn the combustion efficiency of the fires. However, while both $CH_4$ and CO emission factors, relative to burnt area or $CO_2$, are expected to increase in response to a reduction in combustion efficiency[34,35], there is currently no established relationship between combustion efficiency and the $CH_4/CO$ ratios. To the best of our knowledge, measurements

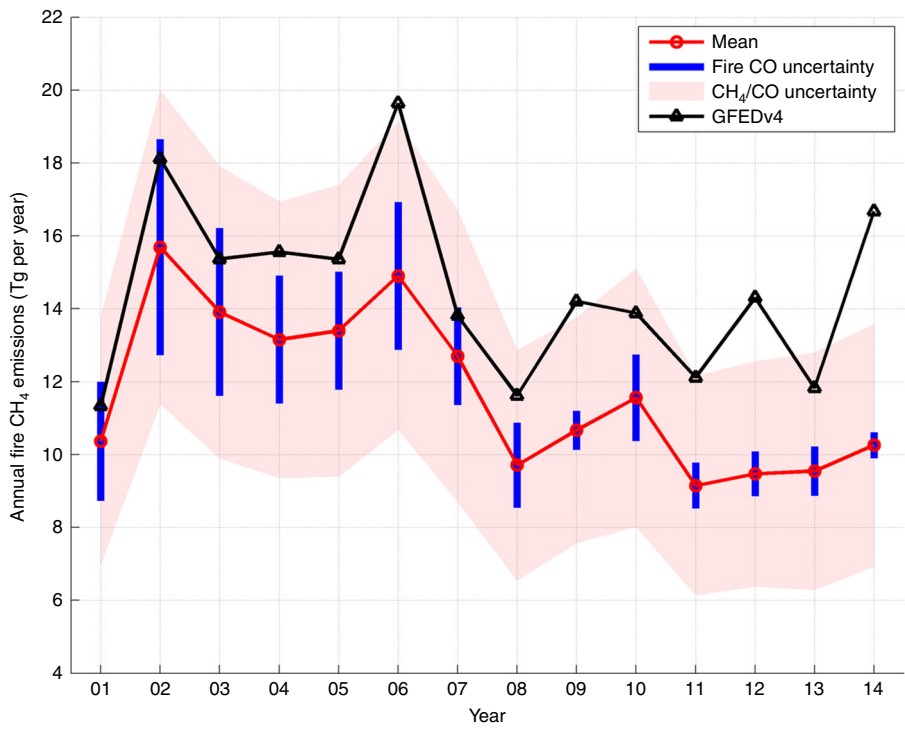

**Fig. 1** Trend of methane emissions from biomass burning. Expected methane emissions from fires based on the Global Fire Emissions Database (black) and the CO emissions plus $CH_4/CO$ ratios shown here (red). The range of uncertainties in blue is due to the calculated errors from the CO emissions estimate and the shaded red describes the range of error from uncertainties in the $CH_4/CO$ emission factors

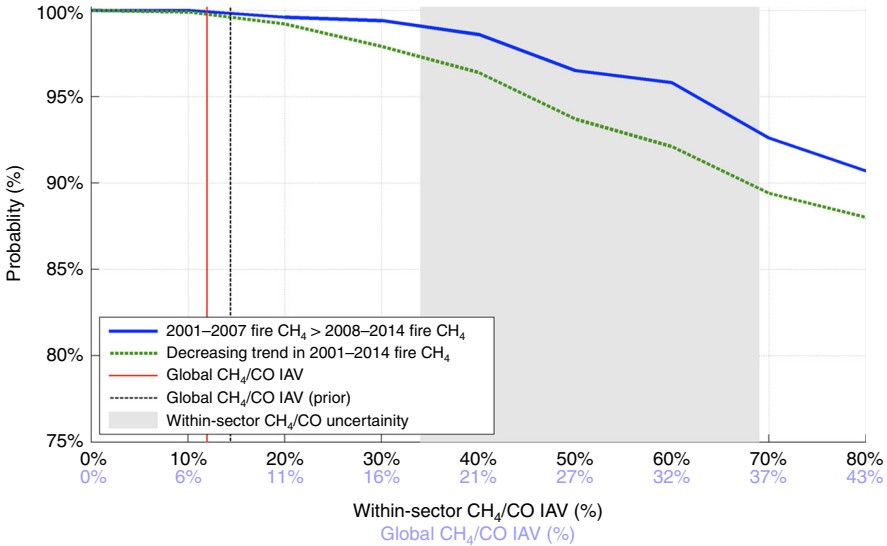

**Fig. 2** The probability of a decrease in biomass burning methane emissions during 2001–2014. Probability of decrease if the emission factors are within-sector $CH_4/CO$ inter-annual variability (black, x axis) and the corresponding global-scale $CH_4/CO$ inter-annual variability (light blue, x axis). The probability estimates include the propagation of systematic errors in fire CO emission estimates, and sector-specific $CH_4/CO$ values. For comparison, the vertical lines show the global $CH_4/CO$ IAV due to annual changes in relative fire sector contributions. The gray-shaded area shows the within-sector $CH_4/CO$ uncertainty

tracking temporal changes in fire CH$_4$/CO ratios indicate no coherent relationship between fire phase and CH$_4$/CO variability on daily timescales[34,35] or any significant relationship between seasonal CH$_4$/CO variability and combustion completeness[36]. Ultimately, joint constraints on the temporal variability of CH$_4$/CO, e.g., based on further in situ monitoring of fire CH$_4$/CO or

**Table 1 Isotopic signatures of the three source categories used in our box-model analysis**

| Source type | Previous literature δ$^{13}$C-CH$_4$ (‰) | Schwietzke et al._2016[15] δ$^{13}$C-CH$_4$ (‰) |
|---|---|---|
| Biogenic | −60 ± 4.3 | −62.3 ± 0.7 |
| Fossil fuel (+natural seepages) | −39 ± 1.7 | −44 ± 0.7 |
| Biomass burning | −24.0 ± 2.0 | −22.3 ± 1.9 |

The isotopic signatures are reported as means ± 1-sigma uncertainty

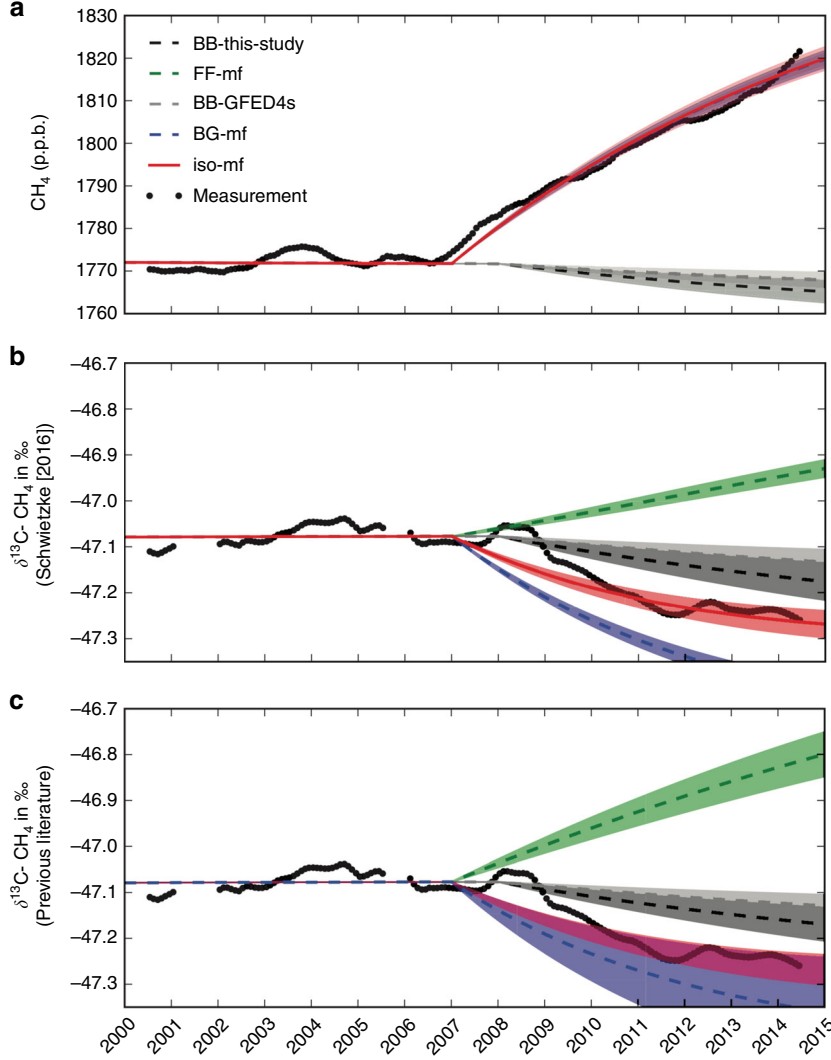

**Fig. 3** Simulated CH$_4$ and δ$^{13}$C-CH$_4$ values. CH$_4$ mixing ratios (**a**) and simulated by the box model for values shown in Table 1 and listed by model scenario in Table 2. **b**, **c** describe the simulated δ$^{13}$C-CH$_4$ using the updated values from Schwietzke et al.[15] and from prior literature. The biomass burning changes are prescribed based on the estimates from this study (BB-this-study) and GFED4s (BB-GFED4s). For the BG-mf and FF-mf scenarios, the CH$_4$ mole fractions growth is explained by an emission increase of only biogenic or only fossil fuel, respectively (BG-mf and FF-mf overlap in **a**). The iso-mf scenario shows the best fit to the isotope and mole fraction data, using an additional source of 24.7 ± 1.4 Tg CH$_4$ per year with an isotopic signature of −56.1 ± 1.1‰. The required adjustments to the methane budgets for fossil fuel and biogenic sources are shown in Figs. 4 and 5. The 1-sigma error margins are the propagated uncertainties of isotopic source signatures and uncertainties of the perturbations. The measurements shown here are the calculated global average of NOAA-ESRL network measurements

**Table 2 Description of CH$_4$ box-model scenarios**

| Scenario name | Constrained by CH$_4$ | Constrained by δ$^{13}$C | Biomass burning change in 2008 | FF and BG change in 2007 | OH change |
|---|---|---|---|---|---|
| BB-this-study | No | No | This study | No change | No |
| BB-GFEDv4s | No | No | GFEDv4s | No change | No |
| BG-mf | Yes | No | No change | Only BG increase | No |
| FF-mf | Yes | No | No change | Only FF increase | No |
| Iso-mf | Yes | Yes | Three BB change scenarios[a] | Constrained[a] | No |
| Iso-mf-OH | Yes | Yes | Three BB change scenarios[a] | Constrained[a] | 0–3% reduction |

These scenarios are presented in Figs. 3, 4, 5
[a] Multiple scenarios are derived based on three BB change scenarios (this study, GFEDv4s, and no change), where BB and BG are constrained based on CH$_4$ δ$^{13}$C source signatures and their associated uncertainties (Figs. 4 and 5)

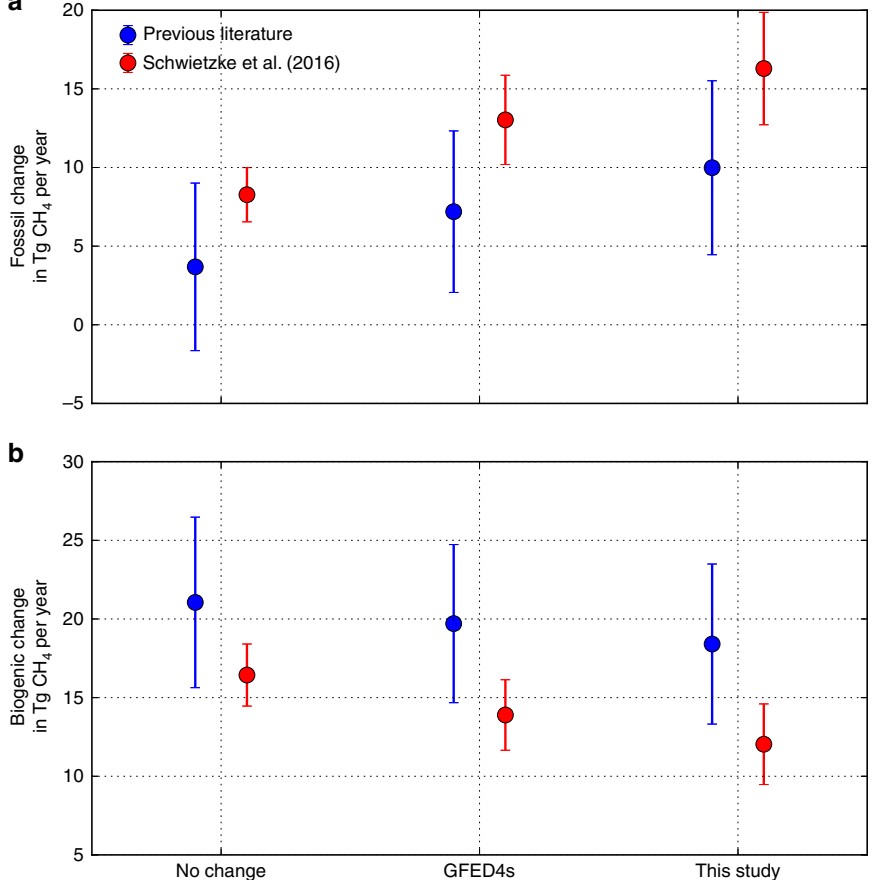

**Fig. 4** Change in average annual biogenic and fossil fuel emissions. Change in average annual fossil fuel (**a**) and biogenic (**b**) emissions between the 2001–2006 and 2007–2014 periods needed to fit the CH$_4$ mole fraction for different assumptions about biomass burning emissions and the isotopic signatures of the methane emission sources. These values are calculated for different proposed changes in biomass burning emissions: GFED4s = 2.1 ± 1.1 Tg CH$_4$ per year, this study = 3.7 ± 1.4 Tg CH$_4$ per year, and no change = 0.0 Tg CH$_4$ per year. The isotopic signatures assigned to each source type are shown in Table 1. The error bars are the 1σ uncertainties, which are calculated by propagating the uncertainties of the source isotopic signatures, biomass burning perturbations, and total perturbations needed to fit the growth rate and isotope measurements (see iso-mf scenario in Fig. 3)

CH$_4$ and CO column measurements from upcoming TROPOMI satellite mission[37], could be key to improving the accuracy of fire methane emission estimates derived from atmospheric CO constraints.

**Balancing the isotopic budget of methane**. When accounting for a 3.7 ± 1.4 Tg CH$_4$ per year decrease in biomass burning emissions between 2001–2007, and 2008–2014, the net change of other components of the methane budget (e.g., FF, BG sources, or a change in the OH sink) must have been even stronger than previously assumed in order to explain the observed increase in global atmospheric CH$_4$ levels. Based on our estimated reduction in biomass burning emissions, we quantify the contribution of FF-related and BG sources to atmospheric methane increase using ground-based measurements from the National Oceanic and Atmospheric Administration Earth System Research Laboratory (NOAA/ESRL) network (Methods). Relative to FF-related sources, methane from BG sources is generally depleted in $^{13}$C, while methane emitted by biomass burning is relatively

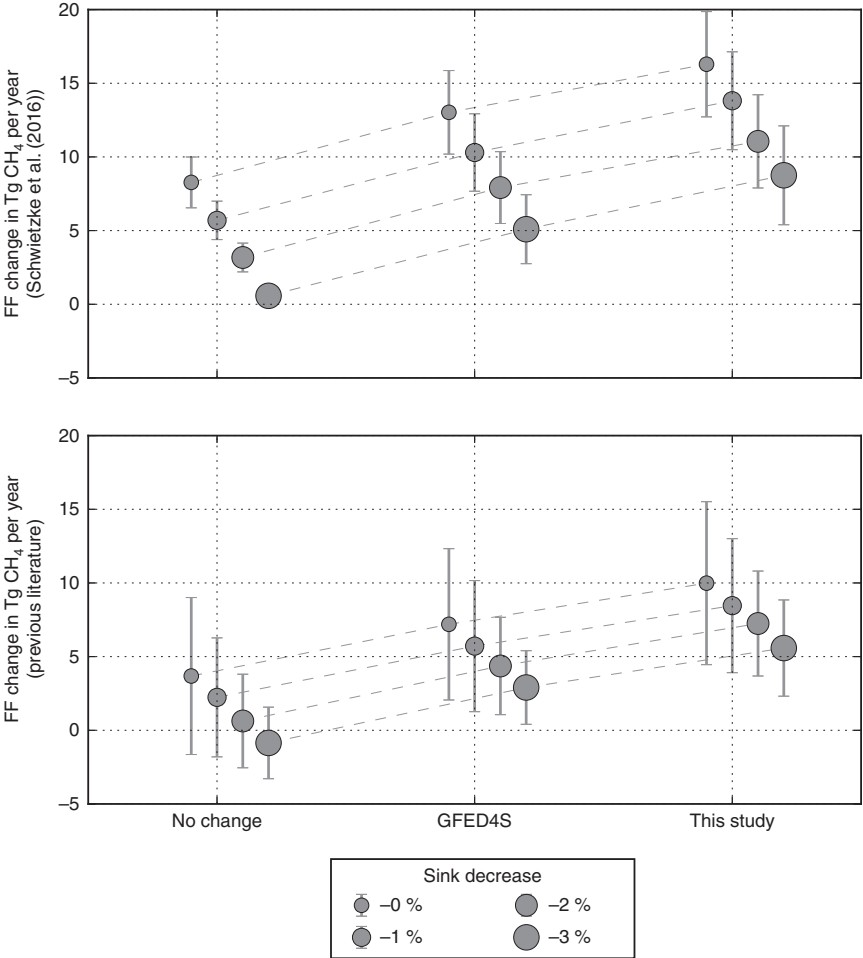

**Fig. 5** Fossil fuel change needed to fit the observed $CH_4$ growth rate and isotopic composition. Fossil fuel change needed to fit the observed $CH_4$ growth rate and isotopic composition assuming a simultaneous change in the $CH_4$ lifetime due to a change in sink. The results for a constant (0%) sink change correspond to the iso-mf scenario (red lines in Fig. 3)

enriched in $^{13}C$ (Table 1)[5,14,15]. Recent updates for the isotope signatures of these emission categories have profoundly changed the global partitioning between source categories, resulting in a larger FF contribution to atmospheric methane mole fractions[15]. We constrain an atmospheric single box model (Methods) using isotope signatures to estimate the contributions of BG, FFs, and biomass burning methane sources to atmospheric methane[5,38]. We use both new estimates of isotope signatures[15] as well as previously accepted isotope signatures for our methane source partitioning estimates (Table 1) in order to verify that these differences in the isotopic composition do not affect our conclusions.

Figure 3 shows the box-model results for a range of scenarios (Table 2) that could explain the observed increase in the $CH_4$ mole fraction, but would yield different temporal isotope trajectories, under the assumption of a constant or varying atmospheric OH sink during this time period (Methods). Attributing the $CH_4$ mole fraction increase to either BG (BG-mf scenario in Table 2) or to FF (FF-mf scenario in Table 2) emissions leads to $\delta^{13}C$-$CH_4$ trajectories that do not agree with the NOAA/ESRL measurements in the post-2007 period. When optimizing the box-model fluxes in order to fit both the $CH_4$ and $\delta^{13}C$-$CH_4$ time series to the NOAA/ESRL network measurements (iso-mf scenario), the fits correspond to an additional global methane source of $24.7 \pm 1.4$ Tg $CH_4$ per year with average

isotopic signature of $-56.1 \pm 1.1‰$; for the iso-mf scenario, this additional source has been partitioned into contributions from BG and FF source categories for three scenarios of BB emission change: no change in biomass burning; the current GFED4s estimate ($-2.1$ Tg $CH_4$ per year); and our CO-based top-down estimate ($-3.7 \pm 1.4$ Tg $CH_4$ per year).

We find that the larger-than-expected reduction in methane BB emissions ($-3.7 \pm 1.4$ Tg $CH_4$ per year) leads to a substantial shift of the global methane source increase from BG to FF emissions, due to the impact of decreasing $^{13}C$-enriched BB emissions in the $CH_4$ isotope budget (Fig. 4). For both choices of isotopic source signatures used in this study, the required increase in FF emissions is 12–19 Tg $CH_4$ per year with a corresponding increase in BG emissions of 12–16 Tg $CH_4$ per year. As shown in Fig. 4, FF contributions have to become an increasingly larger contribution to the overall increase in methane to account for larger decreases in biomass burning in order to also balance the isotopic budget. The required FF emission enhancement found here is substantially larger than in previous literature[5], which showed a contribution of approximately 5.5 Tg $CH_4$ per year from FF when assuming BB changes of between 0 and $-1.5$ Tg $CH_4$ per year. In principle, a compensating increase in biofuels could cancel the decrease in the biomass burning because their isotopic signatures are similar. However, there are no current measurements of a concurrent biofuel emissions increase and

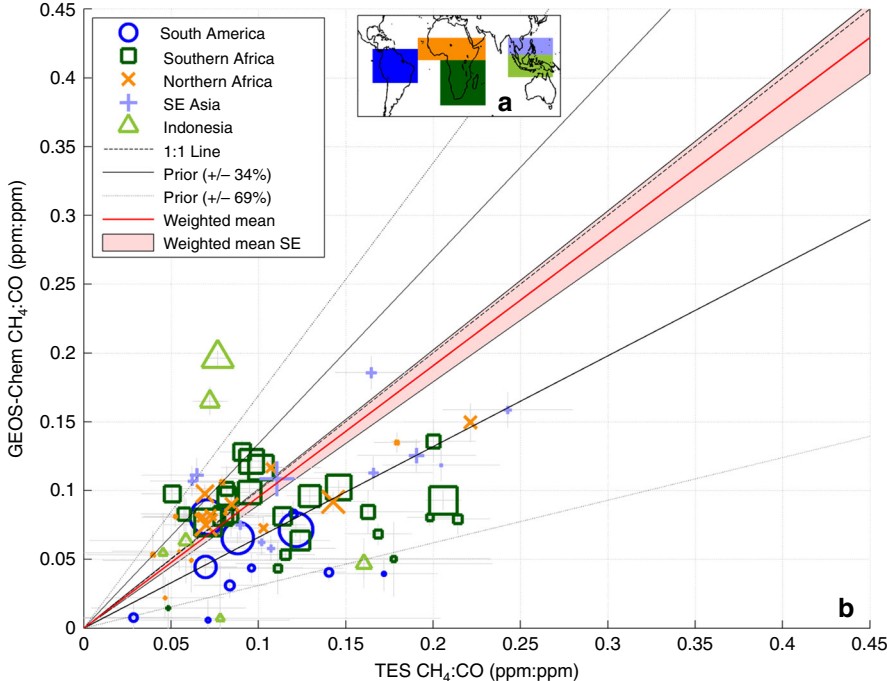

**Fig. 6** Comparison of $CH_4/CO$ ratios from the GEOS-Chem model and Aura TES data. **a** Comparison of $CH_4/CO$ ratios observed in tropical and subtropical fire plumes from the Aura TES data to those expected from the GEOS-Chem model with GFED-based emission factors. **b** The regions corresponding to symbols in **a**. The best fit (weighted to the size of the fire emissions) and the corresponding standard error (standard error or the pink-shaded area in figure) are shown by the red line and shaded area. Fires from different regions are shown as different symbols. The relative size of the fire emissions is indicated by the relative size of the symbols

furthermore such an increase is unlikely as it would amount to 25% of the estimated yearly total for the biofuel emissions[39].

Recent publications have also shown that we cannot rule out a decrease in the chemical sink of methane (reaction with OH) as the cause for the recent increase[18,19]. To address this possibility, we have performed additional box-model simulations where the sink is decreased progressively from 0 to 3%[19] (Fig. 5). The largest effect of assuming changes in the atmospheric OH sink is that the required global $CH_4$ source changes accordingly. For example, a 3% sink decrease would require a net source enhancement of ~8 Tg $CH_4$ per year instead of ~25 Tg $CH_4$ per year. The isotope source signature required to match the observed temporal evolution of $\delta^{13}C$ also changes, from −56 to −61‰. Using the mass balance equation (Eqs. (10)–(12), Methods), the corresponding FF emission contributions have been calculated for the different BB emissions change scenarios (iso-mf-OH scenario; Fig. 5). As shown in Fig. 5, we find that a FF enhancement of 6–12 Tg $CH_4$ per year is still needed to explain the $\delta^{13}C$ measurements in case of a 3% OH sink decrease; this amount reflects the total excess of ~8 Tg $CH_4$ per year, a 1–6 Tg $CH_4$ per year contribution from BG sources, and the 2.4–5.1 Tg $CH_4$ per year decrease from fires. Therefore, our conclusion that an increase in post-2007 FF emissions is needed to explain the observed shift in methane emissions[5] remains valid, even if a sizeable fraction of the atmospheric methane concentration increase is due to decreasing atmospheric OH concentrations.

In conclusion, this study provides an updated estimate to global emissions of methane from fires that are on the low-end of previous estimates (12.9 ± 3.3 Tg $CH_4$ per year, in contrast to prior estimates of 14–26 Tg $CH_4$ per year[20,21]) for the 2001–2014 time period. We also find that methane emissions from fires decreased after 2007 by 3.7 ± 1.4 Tg $CH_4$ per year; this decrease is substantially larger than the GFED4s estimated reduction (2.1 Tg $CH_4$ per year). Because fire emissions are isotopically heavier than those from FF or BG $CH_4$ sources, the larger-than-expected decrease in fire emissions requires a substantial re-balancing of sources to explain both the recent increase in the mole fraction and isotopic composition of atmospheric methane. We show that new estimates for biomass burning and revisions to the isotopic composition of methane sources[5,15] lead to a revised estimate of the FF and BG contributions to the post-2007 atmospheric methane budget (increase of 12–19 Tg $CH_4$ year and 12–16 Tg $CH_4$ year, respectively), assuming no change in the atmospheric OH sink of methane; reducing the sink by up to 3% reduces the FF and BG emissions changes to 6–12 Tg $CH_4$ year, and 1–6 Tg $CH_4$ year. Our results therefore reconcile the previously conflicting findings on the recent changes to atmospheric methane and its isotopic composition, where isotopic evidence indicated a BG $CH_4$ emission increase, while ethane/methane measurements indicated an increase in FF $CH_4$ emissions.

## Methods

**Approach for characterizing $CH_4$ emissions from fires.** Our approach for quantifying $CH_4$ emissions from fires using satellite-based CO and $CH_4$ concentration measurements is intended to mitigate and characterize uncertainties due to (1) errors in transport and chemistry, (2) uncertainties in partitioning CO emissions on the GEOS-Chem grid back to a priori CO emission types, and (3) uncertainties in the $CH_4/CO$ emission factors and their IAV. As discussed in the following sections, we first quantify monthly CO fluxes and their uncertainties at monthly timescales on a $5 \times 4°$ (longitude × latitude) grid using measurements of CO concentrations from the Terra Measurements of Pollution in the Troposphere (MOPITT) satellite instrument (V6J multi-spectral product[40] and the adjoint version of GEOS-Chem[31]). CO fluxes are then re-partitioned to the CO emission types plus uncertainties on each $5 \times 4°$ grid cell using a Bayesian Markov Chain Monte Carlo approach[25,41] that accounts for the a priori and a posteriori uncertainties of the BB emissions and other CO emissions. Estimates of the $CH_4$ emissions and their uncertainties are then calculated by multiplying BB CO emissions by the GFED-based estimate of each fire-type contribution, the expected $CH_4/CO$ emission factors for all fire types within each grid cell, and the uncertainties of the GFED-recommended emission factors. The emission factor uncertainties are tested with $CH_4$ and CO measurements from the Aura TES instrument.

**Approach for quantifying CO fluxes.** The approach used to quantify CO fluxes over 15 years using the GEOS-Chem adjoint and Terra MOPITT data is described in previously published research[31]. In summary, the inversion approach is to compare MOPITT data, averaged hourly and on the GEOS-Chem $5 \times 4°$ degree grid, to the model and modified by prior knowledge of CO emissions based on published inventories. The prior error for the CO fluxes on each grid is assumed to 50% and is uncorrelated between grid cells[31]. The error prescribed for each set of hourly, $5 \times 4°$ degree averaged data is 20%, consistent with the mean uncertainty of the MOPITT data. Emissions for the prior CO fluxes are also averaged onto the GEOS-Chem grid. As discussed in previous studies[42,43], observations or models that are coarser than the scales of the actual smoke plumes can have larger uncertainty because of the effects of sub-grid scale diffusion, transport processes, and chemistry. However, the emissions from models at different spatial resolutions that are observationally constrained by satellite concentration data become consistent when averaged over several of the coarser scale model grid cells because the different model posterior concentrations have to be consistent with the observed CO concentrations[42]. The emissions results presented here should therefore be conservatively interpreted as averages of all fire emissions over a month for aggregates of the GEOS-Chem grid cells (~2000 km spatial scales).

The approach for calculating CO emissions mitigates and characterizes error in atmospheric transport and chemistry because they are typically the largest errors when quantifying CO emissions using concentration data[44–46]. For example, errors in the modeled CO fields can be amplified as CO is advected away from a source region due to the accumulation of transport and chemistry errors. We use a two-step approach to reduce the impact of these errors: firstly, we assimilate the MOPITT CO measurements over the ocean so that the modeled CO concentration fields that are advected over land from the ocean are consistent with the satellite data[42]. We then estimate the CO emissions through comparison of model and data just over continental regions. Effectively this approach accounts for advection of the observed CO fields over the continents from the oceans while reducing the sensitivity of emissions from one continent to those from other continents[42].

Our inversion approach reduces, but does not remove, chemistry and transport errors contributions from our CO flux estimates. In order to characterize the remaining CO flux estimate errors, we produce three different estimates that are, respectively, based on the MOPITT CO total column, profile, and lower-troposphere[28,46] concentration measurements. The three concentration measurements have different sensitivities to CO as a function of altitude, and therefore impose varying effects of transport and chemistry errors onto the model concentrations[28,46] after they are passed through the corresponding instrument operators described by the a priori and averaging kernels. For example, estimates based on the total column data will be less sensitive to convection errors because the total column of the model estimate is effectively the same for all ranges of convection. However, these estimates will be the most sensitive to errors in advection and chemistry because the model has to balance these errors with remotely advected emissions. Total column measurements are also less sensitive to nearby surface emissions because the total column is representative of air parcels that originate from hundreds to thousands of kilometers away from the measurements[28]. Similarly, estimates based on the profile data will be more sensitive to emissions near to the measurement site than the total column data but are also more sensitive to errors in convection in the model. Estimates using the lower-tropospheric (lowest three to four levels of the MOPITT CO profile) will be more sensitive to nearby emissions but also more sensitive to errors in convection[28,31,46].

We have increased/decreased confidence in the magnitude and trend of emissions that are similar/different between these three estimates. For example, the largest differences between the three estimates occur in India and Indonesia, regions where there are relatively large emissions and relatively large convective mass fluxes[46], and contributions from remote sources due to strong advection[47,48]. The mean of these three estimates is used for estimating the CO fluxes at each grid box and the variance between the three estimates is used as our uncertainty for these estimates[28,31]. To obtain the uncertainty of the fire emissions of CO, we next need to account for this posterior uncertainty in the CO estimate along with the partitioning of CO to its different sectors (e.g., biomass burning, FFs, and so on) and its uncertainties, as discussed next.

**Partitioning of posterior CO fluxes to CO emission sectors.** In order to partition CO fluxes estimated on the GEOS-Chem grid cells to their corresponding emissions, we use a Markov Chain Monte Carlo approach[25,41]. This approach quantifies the sectoral CO emissions and their uncertainties on each grid cell such that the sum of the emissions and their uncertainties statistically represents the posterior total CO fluxes and its associated uncertainty. In particular, we estimate emissions for biomass burning (BB, including biofuels), FF, and BG sources. For each timestep and grid cell, the vector $\mathbf{x}$ represents the emissions for each CO sector ($\mathbf{x} = [BB, FF, BG]$); $p(\mathbf{x})$ denotes the prior information on the CO emissions for each sector; and $F$ is a scalar denoting the sum of all CO sector emissions ($F = \Sigma [\mathbf{x}]) \times p(F|\mathbf{A})$ denotes the probability distribution of $F$ given atmospheric inversion constraints (denoted collectively as $\mathbf{A}$), which can be expressed via Bayesian inference as

$$p(F|A) \propto p(A|F)p(F). \tag{1}$$

As discussed previously, $p(F)$ is prescribed as a Gaussian distribution with mean equal to the sum of total prior fluxes ($BB_0$, $FF_0$, and $BG_0$) and a standard deviation of $\pm 50\%$[31]. For each grid cell, we model the posterior probability distribution, $p(F|\mathbf{A})$, based on the flux estimates of the three inversion results, $\boldsymbol{f} = [f_1, f_2, f_3]$, where

$$p(F|A) = \bar{\mathbf{f}} \pm \text{StDev}(\mathbf{f}). \tag{2}$$

Similarly, the probability distribution of $\mathbf{x}$ given atmospheric data $A$, $p(\mathbf{x}|A)$, can be expressed via Bayesian inference as

$$p(\mathbf{x}|A) \propto p(A|\mathbf{x})p(\mathbf{x}). \tag{3}$$

The analytical link between $p(\mathbf{x}|A)$ and known distributions $p(F|A)$, $p(F)$, and $p(\mathbf{x})$ is given by joint probability distribution of $\mathbf{x}$, $A$, $F$, $p(\mathbf{x}, A, F)$, where—through the probability chain rule:

$$p(\mathbf{x}, A, F) = p(\mathbf{x}|A, F)p(F|A)p(A), \tag{4}$$

$$p(\mathbf{x}, A, F) = p(F|A, \mathbf{x})p(A|\mathbf{x})p(\mathbf{x}). \tag{5}$$

Since $F$ and $\mathbf{x}$ are conditionally independent of $A$, the above can be summarized as

$$p(\mathbf{x}, A, F) \propto p(\mathbf{x}|F)p(F|A), \tag{6}$$

$$p(\mathbf{x}, A, F) \propto p(F|\mathbf{x})p(A|\mathbf{x})p(\mathbf{x}). \tag{7}$$

Since $p(\mathbf{x}|F) \propto \frac{p(F|\mathbf{x})p(\mathbf{x})}{p(F)}$, the above equations can be expressed as the following distribution:

$$p(\mathbf{x}|A) \propto \frac{p(\mathbf{x})p(F|A)}{p(F)}. \tag{8}$$

The distribution of $p(\mathbf{x})$ is defined as normal, uncorrelated distributions for BB, FF, and BG, with means $FF_0$, $BB_0$, and $BG_0$. The prior distribution of BB is constructed based on monthly total CO emissions from the GFEDv4s inventory[29], and uncertainties in the CO emission factor for each fire type (i.e., savannas, agriculture, forests, and so on). Fire CO emission factors and associated uncertainties are based on those reported in the product GFED4s readme file (http://www.globalfiredata.org/data.html). For each $5 \times 4°$ area, we assumed that the CO emission factor errors from different fire types are uncorrelated. We note that prior fire CO emission uncertainties are possibly underestimated as the roles of fuel load and burned area uncertainties are not well known at $5 \times 4°$ scales, and hence not included in our burned area estimates.

Gridded $5 \times 4°$ monthly uncertainty estimates are not readily available for FF and BG sources. We therefore prescribe the FF prior distribution with a mean FF (or $FF_0$) and corresponding uncertainty of $\pm 50\%$, which is consistent with largest country-level uncertainty reported by previous estimates[49]. Similarly, we prescribe a prior BG distribution of $BG_0 \pm 50\%$. The prior distribution for the CO emissions used in our analysis[31], $p(F)$, is based on GFED version 3, whereas we use GFED4s for our results described here. Due to computational limitations, we are unable to repeat the full 15-year CO inversion used in our analysis with GFED4s. However, the role of the grid cell level GFED version 3 prior is mitigated because the posterior flux distribution $p(F|A)$ is (a) normalized by the GFED3-based prior CO emission distribution $p(F)$, and re-weighed by the GFEDv4s-based prior $p(\mathbf{x})$ using Eq. (8). In addition, the difference between the posterior CO emissions from the prior are typically comparable or larger to the GFEDv3-GFEDv4s difference. We would therefore expect the re-partitioning to provide a similar estimate for the mean CO emissions (within the calculated uncertainties) for the reported time period and have effectively no impact on our conclusions about the trend estimate.

We use an adaptive Metropolis–Hastings Markov Chain Monte Carlo (MHMCMC) approach to sample $p(\mathbf{x}|A)$[50]. Finally, we model the spatial and temporal error co-variances of BB based on total emission estimates $\mathbf{f}$ to match the mean and standard deviations of retrieved BB emissions. For each monthly $5 \times 4°$ retrieval of BB, we create three realizations of BB, $\mathbf{B} = [B_1, B_2, B_3]$ based on the three CO emissions estimates. $\mathbf{B}$ is derived based on the three total inverse CO emissions estimates $\mathbf{f} = [f_1, f_2, f_3]$ as follows:

$$\mathbf{B} = (\mathbf{f} - \bar{\mathbf{f}})\frac{\text{StDev}(\mathbf{b})}{\text{StDev}(\mathbf{f})} + \bar{\mathbf{b}}, \tag{9}$$

where $\bar{\mathbf{b}}$ and StDev ($\mathbf{b}$) represents the retrieved mean and standard deviation of BB within each monthly $5 \times 4°$ grid cell. In this manner, we simultaneously conserve grid scale BB variances while representing a first-order approximation of the BB spatial and temporal error covariance structure.

**Relating estimated CO fire emissions to CH4 fire emissions.** In order to quantify fire $CH_4$ emissions, gridded CO emissions are partitioned into fire types, fuel load, and burned area extent as reported by GFED4s[29] fire emission estimates as discussed in the last section. $CH_4$ emissions and their associated uncertainties (pink-shaded area in Fig. 1) are then derived by multiplying individual fire sector

CO emissions by the mean and standard deviations of sector-specific $CH_4/CO$ emission ratios based on GFEDv4s recommended $CH_4$ and CO emission factors and associated uncertainties. The GFED4s $CH_4/CO$ emission factor is assumed to be constant but with their reported uncertainties varying between 34 and 69%. As discussed in the main text, it is possible that this emission factor could change with different fire phases and combustion efficiency[34–36]. Since these assumptions are based on sparse in situ measurements, we further test the GFED4s $CH_4/CO$ emission factors, and the corresponding 34–69% uncertainty range, with satellite measurements of $CH_4$ and CO in the free troposphere by the Aura tropospheric emission spectrometer (TES) over tropical fires[51] as shown in Fig. 6 and Supplementary Figs. 1, 2, 3, 4. Only TES data that are associated with biomass burning in which CO in the free troposphere is larger than 80 p.p.b. are used because these data are most likely affected by fire emissions[25,28,51]. Transport effects in these comparisons are also mitigated by comparing observed $CH_4/CO$ ratios from the satellite data to those corresponding to air parcels modeled by GEOS-Chem after the model atmospheric concentrations have been convolved with the Aura TES $CH_4$ and CO averaging kernels and a priori constraints in order to account for the vertical resolution and inversion regularization of the Aura TES $CH_4$ and CO estimates[25,28,51]. We find that modeled and TES-observed $CH_4/CO$ ratios are consistent (shaded red area overlaps the one-to-one dashed line in Fig. 6) if these ratios are within ±34% of the GFED4s values (Supplementary Figs. 1 and 2), whereas the model-observation agreement are inconsistent for a −69 and +69% change in $CH_4/CO$ ratios (Supplementary Figs. 3 and 4). Based on these comparisons, we conservatively assume the GFED4 $CH_4/CO$ ratios and their reported uncertainties ranging from 34 to 69% for our analysis.

**Testing a decrease in methane emissions from fires**. The global $CH_4/CO$ ratio uncertainty (pink-shaded area in Fig. 1) is derived as a function of fire sector $CH_4$ contributions and their associated $CH_4/CO$ uncertainties. $CH_4$ emission uncertainty stemming from CO emission uncertainty (Fig. 1, blue bars), is based on the three CO BB realizations discussed previously. We find the fire $CH_4/CO$ emission factor uncertainties are larger than CO-based emission uncertainties and comparable to the resulting trend in $CH_4$ emissions (Fig. 1). Assuming no inter-annual $CH_4/CO$ emission factor variability for each sector throughout 2001–2014, we find a significant decreasing trend in methane emissions (Fig. 2). To test the sensitivity of our result to this assumption about yearly variations in the global mean $CH_4/CO$ emission factor variability, we statistically evaluate the decreasing BB $CH_4$ emission hypothesis under increasing levels of random $CH_4/CO$ IAV within each fire sector. The statistical evaluation is performed by randomly sampling one of the three CO BB emission realizations, their time-invariant $CH_4/CO$ values for each sector based on sector-specific $CH_4/CO$ uncertainty estimates, and their annually varying within-sector $CH_4/CO$ anomalies. We find that the probability of a 2001–2014 $CH_4$ emissions decrease is >95% assuming that sector-specific $CH_4/CO$ IAV is ≤40% of the derived $CH_4/CO$ uncertainty. To our knowledge, $CH_4/CO$ IAV remains poorly characterized, and we cannot reject a $CH_4/CO$ IAV of >40%; however, the corresponding global $CH_4/CO$ variability (~21%) is roughly a factor of 3 greater than the prior and posterior sector-based global $CH_4/CO$ IAV. Moreover, a <95% probability of an emission decrease is only possible when the within-sector $CH_4/CO$ IAV is comparable to the sector $CH_4/CO$ uncertainty (gray-shaded area in Fig. 2). This analysis demonstrates an increasing 2001–2014 $CH_4$ fire trend is only possible if the global fire $CH_4$ emission variability is largely dominated by random global-scale $CH_4/CO$ IAV (i.e., unaccounted by sector-based contributions to global $CH_4/CO$ IAV), and globally coherent inter-annual within-sector $CH_4/CO$ variability is comparable to sector $CH_4/CO$ uncertainty; we note that both statements are theoretically possible but exceedingly unlikely, as these levels of $CH_4/CO$ IAV would be statistically represented with substantially larger uncertainties in sector-specific $CH_4/CO$ estimates.

**Impact of decreasing biomass burning for global $CH_4$ budgets**. The impact of the observed drop in fire emissions on global $CH_4$ is studied using a single box model[5,38], which simulates the $\delta^{13}C$-$CH_4$ values and $CH_4$ mixing ratios, assuming a well-mixed atmosphere. The model uses the following scalar mass balance equations:

$$c(t + \Delta t) = c(t) + [[e_{FF} + e_{BG} + e_{BB}]/m - k \times c(t)] \times \Delta t, \quad (10)$$

$$^{13}c(t + \Delta t) = {}^{13}c(t) + \left[[q_{FF} \times e_{FF} + q_{BG} \times e_{BG} + q_{BB} \times e_{BB}]/m - \alpha \times k \times {}^{13}c(t)\right] \times \Delta t, \quad (11)$$

$$\delta^{13}(t) = \left(\frac{\frac{{}^{13}c(t)}{c(t) - {}^{13}c(t)}}{r_{std}} - 1\right) \times 1000\%, \quad (12)$$

where $c(t)$ and $^{13}c(t)$ are global mean mixing ratios of $CH_4$ and $^{13}CH_4$ at time $t$, respectively. $e_{FF}$, $e_{BG}$, and $e_{BB}$ are methane emissions from FF (thermogenic), BG (microbial), and biomass burning (pyrogenic: biomass and biofuel burning) sources, respectively. $m$ is a factor of 2.767 Tg $CH_4$ per p.p.b. used to convert emissions into atmospheric mole fractions. $k = \frac{1}{\tau}$ is the first-order removal rate

coefficient, where $\tau$ (=9.1 years) is the atmospheric lifetime and $\alpha = \varepsilon + 1$, where $\varepsilon (= -6.8\%)$ is sink-weighted isotopic fractionation of the $CH_4$ in the atmosphere[5]. $q_{FF}$, $q_{BG}$, and $q_{BB}$ are the $^{13}CH_4$ fractions of the corresponding emissions. The model is numerically discretized to run at daily resolution ($\Delta t = 1$ d). $\delta^{13}(t)$ is the global mean $\delta^{13}C$-$CH_4$ value at time $t$.

We assume that $CH_4$ mixing ratios are in a steady state between 2000–2006 and invert global emissions to optimize the agreement between the model and the mole fraction and isotope measurements from 2007 onwards. Although we report the decrease in BB emissions for the time periods between 2001–2007 and 2008–2014, we choose the year 2007 as our start year for the flux inversion because it provides the most realistic fit in our highly simplified model setup. The isotopic source signatures[5,15] used in the model are listed in Table 1. We perform 1000 Monte Carlo simulations for each scenario to account for the uncertainties in the isotopic source signatures and the associated emission adjustments. All the associated uncertainties are assumed to be normally distributed. For each run, a randomized isotopic source signature is selected based on the isotopic source signature uncertainty distribution. For all runs, the pyrogenic contribution (biomass burning + biofuel burning) to the total annual methane source is fixed to 35 Tg $CH_4$ per year[20,21] for the period with no biomass burning perturbation (i.e., 2001–2007). The FF and BG contributions are adjusted to match the mean values of $CH_4$ and $\delta^{13}C$-$CH_4$ between 2000–2007. Each of the different scenarios shown in Fig. 4 is constrained to fit the observed growth from beginning of 2007 until the end of 2014. The biomass burning perturbation starts in 2008, corresponding to the transitioning between periods of higher and lower biomass burning in Fig. 1.

Biogenic and FF emission perturbations are introduced in 2007 to fit the observed $CH_4$ mole fraction increase. To determine these numbers, we select NOAA-ESRL sites with both $CH_4$ mole fraction and isotope measurements. Only sites with a minimum of 2 years of data between 2001–007 were selected, so that the corresponding steady state was well defined. From these data, a global mean time series is derived for $CH_4$ and $\delta^{13}C$-$CH_4$ (Supplementary Fig. 6)[52]. Thereafter, the mix of sources in the box model is adjusted to find the source composition compatible with the global mean steady state. Then, the box model is run with perturbations within a range of realistic emission increases (+10 to 40 Tg $CH_4$ per year) and isotopic signatures (−45 to −70‰). We select the emission strength and isotopic signatures with the minimum root mean square deviation (RMSD) between model and measurements.

**Choice of 2007 as start of the emission perturbation**. Biogenic and FF emission perturbations are introduced in 2007 to fit the observed $CH_4$ mole fraction increase. This choice was made after comparing the RMSD between the measurements and the best fit case when starting the optimization in different years (Supplementary Table 1). 2007 was selected as it resulted in the lowest RMSD. Supplementary Table 1 shows the goodness of fit as measured by the RMSD between $CH_4$ mole fractions measurements and optimized box-model simulations for starting years varying between 2005 and 2008. The corresponding strength of the optimized methane emission perturbation is also given.

**Calculation of global average $CH_4$ and $\delta^{13}C$-$CH_4$**. Here we describe the method used to calculate a global representative time series shown in Fig. 3 of the main text. We use only stations with a sufficient number of measurements for both $CH_4$ and $\delta^{13}C$-$CH_4$, so that our model is fit to measurements representing the same air masses. In the first step, zonal averages are taken of measurements in four latitudinal bands: NET (Northern Extra Tropics: 30°N–90°N), NTRO (Northern Tropics: 0°–30°N), STRO (Southern Tropics: 30°S–0°), and SET (Southern Extra Tropics: 90°S–30°S). The time series for each station is shown in Supplementary Fig. 5 and for each zonal average in Supplementary Fig. 6. We derive global representative measurements by averaging the time series for each zone. The zones are weighted equally in the averaging as they represent the same area and hence approximately the same air mass. This method avoids the problem that a disproportionally large number of stations in one particularly zone biases the global mean. Note that in the zonal to global averaging, we use months when zonal means are available for each zone. For example, we skipped a few months of 2001, as isotopic values were unavailable for NET. This approach minimizes the influence of a varying representation of zones due to limitations in measurement availability. Using this method, we find an optimum fit to the $CH_4$ and $\delta^{13}C$-$CH_4$ data when adding 25.7 ± 1.4 Tg $CH_4$ per year with an isotopic signature of −56.1 ± 1.1. A recent study[5] performed a similar analysis but obtained a slightly different average additional source strength of 19.7 Tg $CH_4$ per year and a range of −56 to −61‰ for the isotopic composition. The disagreement between their values and our best fit scenario can be explained because they fit their isotope model for the 2006–2014 time period, whereas we start at 2007 (see previous section) and because they use a different set of NOAA sites to calculate the global mean $CH_4$ and $\delta^{13}C$-$CH_4$ time series.

**Data availability**. All data used here are publicly available through the Terra MOPITT website, https://www2.acom.ucar.edu/mopitt and the NASA AVDC repository, https://eosweb.larc.nasa.gov/project/tes/l2_lite_table. Atmospheric $CH_4$ mole fraction and $\delta^{13}C$-$CH_4$ data are publically available through NOAA GMD website, www.esrl.noaa.gov/gmd/. Isotopic composition of atmospheric methane

data are taken from: White, J.W.C., Vaughn, B. H. and Michel, S. E. (2017), University of Colorado, Institute of Arctic and Alpine Research (INSTAAR), Stable Isotopic Composition of Atmospheric Methane ($^{13}$C) from the NOAA ESRL Carbon Cycle Cooperative Global Air Sampling Network, 1998-2015, Version: 2017-01-20,Path: ftp://aftp.cmdl.noaa.gov/data/trace_gases/ch4c13/flask/. Surface $CH_4$ data are taken from:Dlugokencky, E.J. et al. (2017), Atmospheric Methane Dry Air Mole Fractions from the NOAA ESRL Carbon Cycle Cooperative Global Air Sampling Network, 1983-2016, Version: 2017-07-28, Path: ftp://aftp.cmdl.noaa.gov/data/trace_gases/ch4/flask/surface/.

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

## Acknowledgements

This research was carried out at the Jet Propulsion Laboratory, California Institute of Technology, under a contract with the National Aeronautics and Space Administration. The work at Utrecht University is related to the program of the Netherlands Earth System Science Centre (NESSC), financially supported by the Ministry of Education, Culture and Science (OCW). The National Center for Atmospheric Research (NCAR) is sponsored by the National Science Foundation. The MOPITT and TES projects are supported by the National Aeronautics and Space Administration (NASA) Earth Observing System (EOS) Program. The MOPITT team also acknowledges support from the Canadian Space Agency (CSA), the Natural Sciences and Engineering Research Council (NSERC) and Environment Canada, along with the contributions of COMDEV and ABB BOMEM. Methane surface data were downloaded from the World Data Centre for Greenhouse Gases. We are very grateful to all the institutions and individuals who provide these surface data for researchers to use as these efforts are critical for carbon cycle science research; the following is hopefully an inclusive list of institutions and individuals, based on email response, who provide data that we use in this research: (1) NOAA, Boulder CO/Ed Dlugokencky, Laboratory for Earth Observations and Analyses, (2) ENEA, Palermo, Italy/Salvatore Piacentino, the CSIRO Flask Network/Paul Krummel, (3) Atmospheric Environment Division, Global Environment and Marine Department Japan Meteorological Agency/Atsushi Takizawa, and (4) Canadian Greenhouse Gas Measurement Program, Environment Canada/Doug Worthy. We would like to thank Dr. David Schimel of JPL for his helpful comments and feedback. We would like to thank the Stable Isotope Lab, CU-INSTAAR: James White, Bruce Vaughn, and Sylvia Michel for use of their data in this analysis.

## Author contributions

J.R.W. led the study. A.A.B. characterized how uncertainties in the total CO emissions and $CH_4/CO$ ratios affected conclusions about the $CH_4$ emission trends. Z.J., T.W.W. and H.M.W. provided CO emissions and supporting analysis using MOPITT data. S.P., S.H., and T.R. led the isotopic analysis.

## Additional information

**Competing interests:** This research was funded through a NASA Carbon Cycle Science ROSES grant NNH13ZDA001N. The remaining authors declare no competing financial interests.

