## [Peer Review File · Nature Communications]

Reviewers' comments:

Reviewer #1 (Remarks to the Author):

Worden et al. examine the causes of recent trends in atmospheric methane mixing ratios as well as its isotopic signature. Specifically, they reconstruct emission rates of methane from biomass burning (BB) from satellite data of burned area and trace gas measurements. The significant finding that BB emissions decreased between a period of stable CH₄ mixing ratios and a renewed increase thereof has the potential to reconcile observed isotope trends with independent evidence that fossil fuel related emissions played a strong part in the increase. This is of interest not only to the scientific community in order to understand the methane budget but also to policy makers in their effort to mitigate climate change. The study is well designed and presented. I have no expertise in some of the statistical methods (Bayesian statistics) underpinning several of the calculations and therefore cannot evaluate them. I have a couple of points for the authors to consider and address, which potentially affect the final numbers and their uncertainties. These are in particular the question of weighting station results in the calculation of global averages and an uncertainty contribution from inter-annual variability in CH₄/CO ratios. Please find details to these points, as well as some comments and suggestions, below. Overall I commend the authors on a valuable contribution to a topic that has seen a lot of interest in the recent literature.

Hinrich Schaefer

Detailed comments

Lines 14-16: this sentence is not quite correct in that “changes in fossil fuels” can explain only one of the mixing ratio and isotope trends but would drive the other in the wrong direction. It is probably worthwhile to be precise here, because the findings of the presented study aim to resolve exactly that contradiction.

Line 16: the findings of reference 3 have recently been contradicted by Bruhwihler et al. (2017), Arguably, the evidence for a role of fossil-fuel methane in the increase that comes from other studies of ethane trends is less controversial (Helmig et al., 2016; Sussmann et al., 2016). The authors could consider these alternative citations in order to avoid a complicated discussion in the abstract and summary.

Lines 21-23: the stated emission rates apply to a time span that includes both the plateau in methane mixing ratio and the renewed increase. It is also not clear when the stated reduction started and what level it is relative to. Further, speaking of a “rate of reduction” seems to imply that the amount of the decrease changes over time. Are the 3.7 Tg/yr the difference between a prior and a later point in time or a rate of reduction per year? It would be helpful to reword this sentence.

Line 37: some of the quoted studies also attribute sink variability, which could therefore also be mentioned here.

Line 40: Did Schwietzke et al. (ref 9) examine drivers of the increase in detail? Nisbet et al. (2016) may be a more appropriate reference here.

Lines 40-42: please see comments above regarding the controversy around fossil fuel emission increases in North America.

Lines 43-46: Personally, I have more confidence in the study by Rigby et al. (2017) on a possible contribution of OH to the CH₄ increase because it provides an assessment of uncertainties. Is it correct to say that these studies don't show a change in OH, since they reconstruct it from methyl chloroform data?

Line 69 and following: the description of results focuses on differences in averages between the two periods; it doesn't seem to mention that a marked change coincides with the onset of the renewed increase. The evidence is arguably not very strong in light of the uncertainty margins. Yet, it provides circumstantial support to the findings that the authors may choose to discuss.

Lines 80-99: If I follow this correctly, the authors demonstrate that uncertainties in temporal CH₄/CO emission rates cannot reverse the finding of a biomass burning emissions decrease. However, they ignore the contribution of these uncertainties to the error margins in the amount of decrease that they present. It seems plausible that CH₄/CO ratios would have changed due to wetter conditions that lead to biomass burning reductions in the first place (I would assume that there would not only be fewer/smaller fires overall but that there would be more smouldering). Therefore, the error margins of the emission reductions that inform the analysis of overall changes in the methane source mix (such as the isotope mass balances) should include the contribution from CH₄/CO IAV. It could be discussed if there are mechanistic reasons that the impact on uncertainties may not be stochastic but directional. Relevant to this point is that so far the manuscript provides little context for the meteorological changes that underlie the postulated emission changes, notably the predominance of La Nina conditions with strong events in the 2007-2014 period.

Line 112: is reference 17 the correct one? Should it be ref 9?

Lines 112-114: testing the scenarios against the previously assumed, i.e. outdated isotope signatures makes for a more robust case. Having said that, if the authors find themselves pressed for space this could be done as a sensitivity test and presented in the supplement. This is a suggestion only.

Line 114-122 (and Fig. 3 caption): it should be made clear that Fig. 3 shows only one example of an ensemble of model runs for various station records. Also, strictly speaking the red lines in Fig. 3 correspond to the specific values derived for Cape Tutuila and not to the global averages quoted here (unless I am mistaken). In this paragraph, it would be helpful, if not necessary, to explain that the analysis includes a globally representative number of stations. As described here I misunderstood the design and had to read the supplementary material. The reader should be able to understand the basic design from the main text.

Line 134: Personally, I find percentages easier to grasp than ratios, however, that may be an individual preference.

Lines 140-146: the main interest of this study is the finding that fossil fuel emissions contributed a substantial percentage of the recent increase. It should be documented here that this finding holds in light of possible sink changes. A minimum percentage of FF to the increase should be given. Further discussion of the studies of Turner et al. (2017) (Ref. 5) and Rigby et al. (2017), which both appeared in PNAS since submission of this manuscript is needed. Given the large uncertainties in the PNAS studies and the evidence from Nisbet et al. (2016) that OH changes are not consistent with the observed seasonal variabilities, a small minimum increase found here does not necessarily detract from the contribution made by the study presented here.

Lines 161-165: this sentence would benefit from rewording, including the redundant mentioning of the role of sinks.

Line 321: it should be explained how τ and ϵ relate to κ and α , respectively. Otherwise, the reader cannot follow the box model equations.

Line 329: please clarify which source class (thermogenic or pyrogenic) biofuel burning is part of. See contradicting statement in line 320.

Lines 333-334: where is it presented that 2008 is the appropriate start year? I see that average differences are calculated for periods ending 2007 and starting 2008 (instead of, say, 2006 and 2007), but I don't seem to be able to find the justification for that choice.

Fig. 2: how is a detectable increase defined for this test? I would recommend to omit the test and this figure altogether. Instead, the contribution of CH₄/CO IAV should be included in the uncertainty margins of the reduction estimate as discussed above.

Fig. 3: the caption should express clearly that this is only one from several scenarios, which all inform the derived global averages. Is the numbering of references in the caption correct? The quoted best fit values for iso-mf are the global averages derived from all stations. Are the red lines the results for this specific station or for the global estimate?

Supplement:

Supplement A: there a reason that the prior emission estimates use GFED3 whereas other calculations in this study use GFED 4s?

Supplement C: I am not sure that I completely follow this exercise. However, it seems that contrary to the text of the supplement Figs. S1 and S3 show that an increase of the emissions by 34% provides an acceptable fit close to the 1:1 line.

Supplement D: while the presented approach of deriving strength and isotopic composition of the additional source (by fits to individual stations and subsequently averaging those) avoids potential shortcomings of the earlier work by Schaefer et al. (2016), it is not without its own problems. The stations include locations like TAP and WAL that show very strong variability, suggesting that local rather than regional dynamics could be dominant. Given the limited number of stations with $\delta^{13}\text{C}_{\text{CH}_4}$ observations this is probably unavoidable. However, without weighting of the stations for their global representativeness local influences could affect the global mean disproportionately. For mid-latitudes, northern stations (where FF emissions are largest) outweigh southern ones 4:1; for the extra-tropics the N:S ratio is 6:2. Hawaii with two stations is weighted equally to the southern extra-tropics. If my back-of-the-envelope calculations are correct, none of these points will change the main findings because the percentage of fossil fuel methane of the renewed increase remains

between ~40% and 65% for all combinations of source values derived in Table SD1. Nevertheless, the calculations should be revisited to derive the best possible source estimates. For the interested reader, it would also be important to see fits for stations other than Tutuila (SMO), in order to assess whether outliers could influence the resulting fit.

Table SD2: The caption states that the emission rates listed under “Schwietzke et al.” and “previous literature” are FF. However, contrary to the expectation that sink increases will lead to a stronger role of biogenic sources (because a smaller additional source requires more isotopic leverage towards ^{13}C -depletion) the listed values indicate an increase in percentage of the total additional source. Please clarify. Different column headers will probably be helpful.

References (unless listed in presented manuscript):

Bruhwyler, L. M., et al. (2017), U.S. CH₄ emissions from oil and gas production: Have recent large increases been detected?, *J. Geophys. Res. Atmos.*, 122, 4070–4083, doi:10.1002/2016JD026157.

Hausmann, P., Sussmann, R., Smale, D., 2016. Contribution of oil and natural gas production to renewed increase in atmospheric methane (2007-2014): top-down estimate from ethane and methane column observations. *Atmospheric Chemistry and Physics* 16, 3227-3244.

Helmig, D., et al., 2016. Reversal of global atmospheric ethane and propane trends largely due to US oil and natural gas production. *Nat Geosci* 9, 490-495.

Nisbet, E.G., et al., 2016. Rising atmospheric methane: 2007-2014 growth and isotopic shift. *Global Biogeochemical Cycles* 30, 1356-1370.

Reviewer #2 (Remarks to the Author):

The paper by Worden et al. titled “Lower Biomass Burning Emissions Since 2006 Reconcile Methane Budgets Based on CH₄ and CH₄” takes on the question of what caused the recent atmospheric CH₄

increase and the accompanying shift towards lighter $^{13}\text{CH}_4$ values. A prior high-profile publication on this matter (Schaefer et al., Science 2016) suggested the primary driver of the recent increase in atmospheric methane must have been biogenic, given the isotopic signature of the methane increase is a shift towards lighter values. Naturally, this conclusion relied heavily on the then generally accepted isotopic signatures of different methane sources and it seemed to conflict observed increases in fugitive emissions from the booming natural gas and oil industries in North America and Europe. In a recent publication, Schwietzke et al. (Nature, 2016) suggested significant updates to methane isotopic source signatures with lower uncertainties. The most significant suggested change was a 5 per mil shift towards lighter values for the fossil-fuel related sources, accompanied by more modest changes in biogenic (lighter) and pyrogenic (heavier) methane sources.

One important implication of the changes to isotopic signatures of methane sources is that fossil-fuel related sources are now a much weaker isotopic lever (at -44 per mil the mean isotopic signature is now comparable to the atmospheric mean of about -47 per mil) than both biogenic and pyrogenic sources. Consequently, atmospheric $^{13}\text{CH}_4$ measurements are not highly sensitive to changes in fossil-fuel sources of methane. It was quite clear that a new budgetary assessment of the recent methane rise was necessary and the renewed effort had to include independent information about changes in biogenic or biomass burning sources. This work by Worden et al. sets out to quantify and lower the uncertainties in the estimates of pyrogenic methane emissions using fire CO emission estimates as a basis. They also put in considerable effort to better characterize the uncertainties in these emissions. This is the original contribution of this work.

Despite being a relatively minor source of atmospheric methane, fire related emissions with a mean isotopic signature of -22 per mil are a very strong isotopic lever. After identifying a decline in pyrogenic methane emissions that is concurrent with the atmospheric increase in methane, the authors show that the isotopic implications of this decrease in fire emissions is strong enough to hide the isotopic signature of a much larger increase in fossil-fuel related methane emissions that led to the methane rise in the atmosphere. It is likely that the new source isotopic signatures (Schwietzke et al., 2016) alone necessitate a significant contribution from increasing fossil-fuel emissions to the recent methane rise. Existing GFED fire emissions inventory enhances the required contribution from fossil-fuel emissions, and the CO-based fire CH_4 emission estimates presented in this paper push the fossil-fuel contribution to a high enough range that there is now agreement with independent estimates of fugitive emission from North America and Europe (e.g. Turner and Franco et al. 2016).

I find that the methods used in this manuscript are sound and the main findings are convincing, and I do not doubt that the CO-based fire emission estimates will be duly scrutinized and tested by other researchers. It should be reasonably easy to replicate the work on CO fire emissions estimates although some clarifications are needed when it comes to the modeling work and deriving the CH_4/CO emission ratios (see questions and comments in attached document). Despite my general

positive opinion about the substance of the paper, I have serious concerns about the writing in general. I simply did not understand some sentences and this impacted my understanding of some aspects of the analysis. I attached an annotated manuscript that includes comments, suggestions, questions, and some editing and deleting. Regards.

Murat Aydin

Reviewer #3 (Remarks to the Author):

This study addresses the key issue of the attribution of recent increase in CH₄ global concentrations to a specific process. Considering the very extensive amount of work behind this manuscript, the quality of the scientific approach and the conclusive elements brought by the authors, this manuscript must be considered for publication in Nature Communications.

The authors use complementary and robust approaches to quantify the contribution of biomass burning to the global emission trends and update contributions from fossil fuel and biogenic sources according to their new estimate of BB.

However, the current presentation and some missing details require major revisions. The following remarks and comments should be addressed before accepting the manuscript.

General comments:

1. Structure

the efforts by the authors to fulfill Nature Communications' requirements in terms of format and length are acknowledged. However, the final shape of the submitted manuscript is an extremely dense and scattered text, hard to follow with the main text referring to the Methods and Supplement and conversely, with a logic sometime hard to follow by the reader.

All the scientific arguments, details on methods and materials, etc. are currently mostly included in the manuscript, but the authors are asked to reconsider the general structure of their manuscript.

Please note that the text and method length limit is 5000 and 3000 words respectively, while the current manuscript has 2000 and 1800 words respectively, leaving space for a large part of the supplement of 2300 words.

In particular, Supplement B, C and D are key elements of the approach and should be included (at least partly with then reference to the Supplement, especially for the figures) in the Method section. The Supplement A.3 seems to present results and should then be included to the main body of the text.

Further details should be included in the main text to make it self-explaining and to bring additional details. For instance:

- please give more details on the numbers for the different contribution to the recent increase for the reader who do not know all the literature by heart; e.g., l. 76-77, the relative magnitude of the 1.5Tg/y becomes clear only after reading all the rest of the text
- l. 116-117: please add details about the sink uncertainty

2. Impact of the model resolution on the results

The CO inversions are based on GEOS-Chem at 4x5 degrees. Although the inversion systems seems robust and smart adaptations are done to limit chemistry and transport errors, the model resolution is quite different to the native resolution of MOPITT.

Are MOPITT measurements averaged over GEOS-Chem grid cells?

Fire emissions are quite localized and generate thin plumes which are not represented with the 4x5 degrees resolution. Please discuss further the possible limitations due to the resolution.

3. Presentation

- The method section and Supplement should be of the same quality as the main body of the text. Some formulations are clumsy in this auxiliary sections.
- Please make notations in the equations consistent. In Sup. A, Eq. 2, vectors and matrices are not in bold font
- l. 306 of Methods, two times M3 title

Technical comments:

- Table SD2: It would help the reader to present this table as Fig. 4
- Sup. A1: prior emissions are deduced from different years and inventories, please comment the consistency of inventories.
- l. 60 "data fusion approach": this doesn't sound like a standard name, from the reviewer's own knowledge
- l. 69-70: the sentence is clumsy
- In M3, l. 321, where is tau used?
- l. 482-484, Figure 4: please reformulate to have a short description of the "no-change" experiment.

Response to Dr. Hinrich Schaefer (Reviewer 1)

General Comment: "...Please find details to these points, as well as some comments and suggestions, below. Overall I commend the authors on a valuable contribution to a topic that has seen a lot of interest in the recent literature."

Response: We would like to express our deepest thanks for your interest and detailed review of our paper, especially as it was your own paper that drove much of the research described here.

Detailed comments

Comment: Lines 14-16: this sentence is not quite correct in that "changes in fossil fuels" can explain only one of the mixing ratio and isotope trends but would drive the other in the wrong direction. It is probably worthwhile to be precise here, because the findings of the presented study aim to resolve exactly that contradiction.

Response: Agreed. We have reformulated the first two sentences in the abstract and broken it into two parts; this change also addresses a comment from reviewer 2. Here is the revised text in the abstract:

"Several viable but conflicting explanations have been proposed to explain the recent ~8ppb/yr increase in atmospheric methane after 2006, equivalent to a net emissions increase of ~25 TgCH₄/yr. A concurrent increase in atmospheric ethane implicates a fossil source; a concurrent decrease in the heavier isotopes of methane points towards a biogenic source, while other studies propose a decrease in the chemical sink (OH)."

Comment: Line 16: the findings of reference 3 have recently been contradicted by Bruhwiler et al. (2017), Arguably, the evidence for a role of fossil-fuel methane in the increase that comes from other studies of ethane trends is less controversial (Helmig et al., 2016; Sussmann et al., 2016). The authors could consider these alternative citations in order to avoid a complicated discussion in the abstract and summary.

Response: Agreed. We have removed this citation from the abstract to avoid confusion but briefly discuss the Turner and Bruhwiler papers in the introduction. We have also added the suggested references about ethane/methane ratios measurements and their implications throughout the introduction.

Comment: Lines 21-23: the stated emission rates apply to a time span that includes both the plateau in methane mixing ratio and the renewed increase. It is also not clear when the stated reduction started and what level it is relative to. Further, speaking of a "rate of reduction" seems to imply that the amount of the decrease changes over time. Are the 3.7 Tg/yr the difference between a prior and a later point in time or a rate of reduction per year? It would be helpful to reword this sentence.

Response: We have re-worded this sentence in the abstract to “Here we show that biomass burning emissions of methane decreased by 3.7 (+/- 1.4) TgCH₄/yr from the 2001-2007 to the 2008-2014 time periods using satellite measurements of CO and CH₄, nearly twice the decrease expected from prior estimates.” to identify the time periods for which we calculate the average decrease (i.e. the difference between the mean CH₄ emissions in 2008-2014 from the mean in 2001-2007

Comment: Line 37: some of the quoted studies also attribute sink variability, which could therefore also be mentioned here.

Response: We discuss sink variability in the introduction (with reference to the Turner et al 2017 study and also now the Rigby et al. 2017 study) as well as added a more thorough discussion on the role of sink variability on our conclusions (see last paragraph in the main section near **lines 167**). While the other papers may also discuss sink variability we feel it might be confusing to add discussion to that effect in this part of the paper.

Comment: Line 40: Did Schwietzke et al. (ref 9) examine drivers of the increase in detail? Nisbet et al. (2016) may be a more appropriate reference here.

Response: We added the Nisbet et al. reference but still include the Schwietzke et al. result here.

Comment: Lines 40-42: please see comments above regarding the controversy around fossil fuel emission increases in North America.

Response: We have removed the citation to Turner in the abstract for this reason but feel the paper would be incomplete without citing these studies. We therefore include both Turner’s and Bruhwiler’s paper in the introduction to refer to the discussion of these matters using methane concentration data.

Comment: Lines 43-46: Personally, I have more confidence in the study by Rigby et al. (2017) on a possible contribution of OH to the CH₄ increase because it provides an assessment of uncertainties. Is it correct to say that these studies don’t show a change in OH, since they reconstruct it from methyl chloroform data?

Response: We have added the Rigby reference now that the paper is published (we had prior knowledge of the Turner paper but not the Rigby paper). My (albeit conservative) understanding of their results is that we cannot rule out OH as a cause; however neither paper can show OH is actually changing through direct measurements or through methylchloroform measurements or by looking at OH pre-cursors. Effectively, both papers’ conclusions are derived through inference using the uncertainties of the known sources and sinks rather than stating a mechanistic reason for OH variability. We have added language to that effect in the paper near line 45:

“Other studies^{18,19} show that we cannot rule out inter-annual variations in the hydroxyl radical (OH) chemical methane sink as the cause; however, these studies do not directly show changes in atmospheric OH or provide a mechanistic reason for a change.”

Comment: Line 69 and following: the description of results focuses on differences in averages between the two periods; it doesn’t seem to mention that a marked change coincides with the onset of the renewed increase. The evidence is arguably not very strong in light of the

uncertainty margins. Yet, it provides circumstantial support to the findings that the authors may choose to discuss.

Response: We agree with the reviewer that the 2006-2008 decrease is noteworthy as it seems that a lot of things change during this time period (CH₄ increase, change in isotopic composition, change in fire emissions). However, we are hesitant to speculate that these changes are tied together. Instead we emphasize the change in fire emissions by stating near **Line 90**; “*This decrease is largely accounted for by a 2.9 ± 1.2 TgCH₄/yr decrease during 2006-2008, which is primarily attributable to a biomass burning decrease in Indonesia and South America^{28,25,31}.*”

Comment: Lines 80-99: If I follow this correctly, the authors demonstrate that uncertainties in temporal CH₄/CO emission rates cannot reverse the finding of a biomass burning emissions decrease. However, they ignore the contribution of these uncertainties to the error margins in the amount of decrease that they present. It seems plausible that CH₄/CO ratios would have changed due to wetter conditions that lead to biomass burning reductions in the first place (I would assume that there would not only be fewer/smaller fires overall but that there would be more smoldering). Therefore, the error margins of the emission reductions that inform the analysis of overall changes in the methane source mix (such as the isotope mass balances) should include the contribution from CH₄/CO IAV. It could be discussed if there are mechanistic reasons that the impact on uncertainties may not be stochastic but directional. Relevant to this point is that so far the manuscript provides little context for the meteorological changes that underlie the postulated emission changes, notably the predominance of La Nina conditions with strong events in the 2007-2014 period.

Response: We agree with the reviewer that wetter years could coincide with increased level of smoldering, and as a result both CH₄ and CO are expected to increase during smoldering combustion; however, to the best of our knowledge, the change in the CH₄/CO emission factor remains uncertain with respect to changes in combustion efficiency. In-situ measurements tracking daily and seasonal fire CH₄ /CO variability indicate no coherent relationship between fire phase and CH₄ /CO variability on daily timescales (Wooster et al., 2011) or any significant changes in in combustion completeness and CH₄/CO variability (Korontzi et al., 2003).

To address the reviewer’s concern, we have now updated the text (paragraph starting near **lines 118** that includes these comments and references:

“To the best of our knowledge, measurements tracking temporal changes in fire CH₄/CO ratios indicate no coherent relationship between fire phase and CH₄/CO variability on daily timescales^{34,35} or any significant relationship between seasonal CH₄/CO variability and combustion completeness³⁶.”

Comment: Line 112: is reference 17 the correct one? Should it be ref 9?

Response: Yes. Thank you for pointing this out. We have corrected the mistake in the revise manuscript (hopefully we did not add new mis-citations!)

Comment: Lines 112-114: testing the scenarios against the previously assumed, i.e. outdated isotope signatures makes for a more robust case. Having said that, if the authors find themselves pressed for space this could be done as a sensitivity test and presented in the supplement. This is a suggestion only.

Response: The higher word limit of the Nature communication journal allows us to keep the results for the old isotopic signatures in the main text.

Comment: Line 114-122 (and Fig. 3 caption): it should be made clear that Fig. 3 shows only one example of an ensemble of model runs for various station records. Also, strictly speaking the red lines in Fig. 3 correspond to the specific values derived for Cape Tutuila and not to the global averages quoted here (unless I am mistaken). In this paragraph, it would be helpful, if not necessary, to explain that the analysis includes a globally representative number of stations. As described here I misunderstood the design and had to read the supplementary material. The reader should be able to understand the basic design from the main text.

Response: We agree with the reviewer that the use of measurements from one particular station, representative of the global mean, in this figure was unnecessarily confusing to the reader. We have revised the analysis using globally averaged measurements in Figure 3. The intended message of the figure remains the same.

Comment: Line 134: Personally, I find percentages easier to grasp than ratios, however, that may be an individual preference.

Response: We have removed this language and instead state the absolute numbers as described in the figures.

Comment: Lines 140-146: the main interest of this study is the finding that fossil fuel emissions contributed a substantial percentage of the recent increase. It should be documented here that this finding holds in light of possible sink changes. A minimum percentage of FF to the increase should be given. Further discussion of the studies of Turner et al. (2017) (Ref. 5) and Rigby et al. (2017), which both appeared in PNAS since submission of this manuscript is needed. Given the large uncertainties in the PNAS studies and the evidence from Nisbet et al. (2016) that OH changes are not consistent with the observed seasonal variabilities, a small minimum increase found here does not necessarily detract from the contribution made by the study presented here.

Response: We agree, although we would like to add that we think that baselining the BB emissions and trend as well as reconciling the biogenic and fossil fuel emissions are also results that are useful to the carbon cycle community. We have added another paragraph and figure (Figure 5) to address the possible influence of a changing sink, as well as additional references to the Turner and Rigby papers

Comment: Lines 161-165: this sentence would benefit from rewording, including the redundant mentioning of the role of sinks.

Response: We have revised the entire summary to reflect our more extensive analysis on the role of the chemical sink on our conclusions and to address other comments by the reviewer. Please see the updated summary section.

Comment: Line 321: it should be explained how τ and ε relate to κ and α , respectively. Otherwise, the reader cannot follow the box model equations.

Response: We have modified the text accordingly:

“ $k = \frac{1}{\tau}$, where τ (= 9.1 years) is the atmospheric lifetime and $= \varepsilon / 1000 + 1$, where ε (=-6.8 per mil) is sink-weighted isotopic fractionation of CH₄ in the atmosphere.”

Comment:Line 329: please clarify which source class (thermogenic or pyrogenic) biofuel burning is part of. See contradicting statement in line 320.

Response: We clarified that biofuels are pyrogenic in the updated text.

Comment:Lines 333-334: where is it presented that 2008 is the appropriate start year? I see that average differences are calculated for periods ending 2007 and starting 2008 (instead of, say, 2006 and 2007), but I don't seem to be able to find the justification for that choice.

Response: 2007 is taken as perturbation start year for total methane emissions and 2008 is taken start year for BB decrease. We have added a section to the Supplemental to explain the rationale: “**Supplemental B. Choice of 2007 as start of emission perturbation:**

Comment: Fig. 2: how is a detectable increase defined for this test? I would recommend to omit the test and this figure altogether. Instead, the contribution of CH₄/CO IAV should be included in the uncertainty margins of the reduction estimate as discussed above.

The results shown in Figure 2 are key to assessing the statistical robustness of our result. The reviewer's suggestion (i.e. including the contribution of CH₄/CO IAV in the uncertainty margins of the reduction estimate) does not (a) account for systematic errors related to the CO inversion methodology, and (b) systematic errors related to the sector-specific CH₄/CO values. To better highlight the value of testing the probability of a CH₄ decrease, we now clarify these points in the updated figure caption.

Comment: Fig. 3: the caption should express clearly that this is only one from several scenarios, which all inform the derived global averages. Is the numbering of references in the caption correct? The quoted best fit values for iso-mf are the global averages derived from all stations. Are the red lines the results for this specific station or for the global estimate?

Response: To avoid the confusion stated here by the reviewer, we have revised this analysis using globally averaged measurements and have updated figures and numbers in the text accordingly.

Comment: Supplement A: there a reason that the prior emission estimates use GFED3 whereas other calculations in this study use GFED 4s?

Response: Thanks for catching this issue! The studies performed by Jiang et al. (2017) used GFED3 with assumed constant values for the burnt area in the last couple of years as GFED had not been updated when the inversion was being setup and processed (note that these decadal

scale inversions take substantial effort and time to setup, run, and evaluate). Furthermore, we were unable to re-do the Jiang et al, (2015) GEOS-Chem CO inversion estimates using GFEDv4s as Zhe Jiang (my former postdoc) has moved on from his position. We note however that the role of the prior GFEDv3 CO emission used in the GEOS-Chem CO inversion is accounted for in our Bayesian attribution approach using Equation 8 in the revised methods. We have added language specifying how we re-normalize to the GFED4 emissions in the revised methods (Section M1.2 near lines 410) as well as a statement that we cannot re-run the CO inversion due to computational reasons. We have also performed an analysis that shows the posterior CO emissions relative to the prior are typically larger than the GFED3 and GFED4 differences; therefore while re-normalizing likely changes what we report for the mean biomass burning emissions of methane we would still expect that the two estimates would be consistent (within the reported error) and that also the trend estimate would remain effectively un-changed. Language to that effect has also been added in section M1.2 near lines 355:

“Due to computational limitations, we are unable to repeat the full 15 year CO inversion used in our analysis with GFED4s. However, the role of the grid-cell level GFED version 3 prior is mitigated because the posterior flux distribution $p(F|A)$ is (a) normalized by the GFED3-based prior CO emission distribution $p(F)$, and re-weighted by the GFEDv4s-based prior $p(x)$ using Equation 8. In addition, the difference between the posterior CO emissions from the prior are typically comparable or larger to the GFEDv3-GFEDv4s difference. We would therefore expect the re-partitioning to provide a similar estimate for the mean CO emissions (within the calculated uncertainties) for the reported time period and have effectively no impact on our conclusions about the trend estimate.”

We have also removed Supplemental A as the Jiang et al. (2017) paper has now been published and have added to the methods section a more complete description of the Jiang et al. papers (and supporting material) describing the CO inversions.

Comment: Supplement C: I am not sure that I completely follow this exercise. However, it seems that contrary to the text of the supplement Figs. S1 and S3 show that an increase of the emissions by 34% provides an acceptable fit close to the 1:1 line.

Response: The aim of this exercise is to test the GFED-recommended CH₄/CO uncertainty estimates when assessing CO And CH₄ emissions at large (e.g. tropical / subtropical) scales because most of the studies documenting CH₄/CO emission factors occur for small-scale plumes. We have rephrased the first and last sentences within this section (now **Supplemental A**) to clarify this objective. Also, we now correctly indicate that the 1:1 line is statistically consistent with the mean vs modelled CH₄/CO when CH₄ emissions are increased (not decreased) by 34%.

Comment: Supplement D: while the presented approach of deriving strength and isotopic composition of the additional source (by fits to individual stations and subsequently averaging those) avoids potential shortcomings of the earlier work by Schaefer et al. (2016), it is not without its own problems. The stations include locations like TAP and WAL that show very strong variability, suggesting that local rather than regional dynamics could be dominant. Given the limited number of stations with $\delta^{13}\text{CH}_4$ observations this is probably unavoidable. However, without weighting of the stations for their global representativeness local influences could affect

the global mean disproportionately. For mid-latitudes, northern stations (where FF emissions are largest) outweigh southern ones 4:1; for the extra-tropics the N:S ratio is 6:2. Hawaii with two stations is weighted equally to the southern extra-tropics. If my back-of-the-envelope calculations are correct, none of these points will change the main findings because the percentage of fossil fuel methane of the renewed increase remains between ~40% and 65% for all combinations of source values derived in Table SD1. Nevertheless, the calculations should be revisited to derive the best possible source estimates. For the interested reader, it would also be important to see fits for stations other than Tutuila (SMO), in order to assess whether outliers could influence the resulting fit.

Response: We have updated our analysis as suggested by the reviewer. A new supplemental section is added describing the updated method:

“Supplemental C: Calculation of Global average CH₄ and δ¹³C-CH₄”

Comment: Table SD2: The caption states that the emission rates listed under “Schwietzke et al.” and “previous literature” are FF. However, contrary to the expectation that sink increases will lead to a stronger role of biogenic sources (because a smaller additional source requires more isotopic leverage towards ¹³C-depletion) the listed values indicate an increase in percentage of the total additional source. Please clarify. Different column headers will probably be helpful.

Response: In accordance with the suggestion of reviewer 3, this table has been converted into a figure and added to the main text. We hope this avoids the confusion.

Response to comments by Professor Murat Aydin (Reviewer 2)

Comment: General Comment: I find that the methods used in this manuscript are sound and the main findings are convincing, and I do not doubt that the CO-based fire emission estimates will be duly scrutinized and tested by other researchers. It should be reasonably easy to replicate the work on CO fire emissions estimates although some clarifications are needed when it comes to the modeling work and deriving the CH₄/CO emission ratios (see questions and comments in attached document). Despite my general positive opinion about the substance of the paper, I have serious concerns about the writing in general. I simply did not understand some sentences and this impacted my understanding of some aspects of the analysis. I attached an annotated manuscript that includes comments, suggestions, questions, and some editing and deleting. Regards.

Response: We would like to thank you for your time in providing a detailed review as well as the very encouraging support of this research. Most of the comments and changes are within the supplied paper revision. I have copied below your comments from this paper and the corresponding response if it looked like a detailed response was required. If it appeared that only a minor response was required we changed the text accordingly.

Detailed Comments

Comment: (Title) It has always been possible to have a methane budget that explains both the CH₄ and δ¹³CH₄ measurements. Schaefer et al. (2016) did it for the recent methane rise in a very different way than yours. The most important implication of this work is the large contribution it attributes to fossil-fuel related emissions – instead of increasing biogenic emissions - in the ongoing methane rise. I think the title would be more impactful if it reflected this.

Response: The authors have discussed extensively this issue (about 12 titles have been proposed! ☺). We feel that while the fossil fuel contribution is important that the most important result of this paper is that we can reconcile the fossil fuel contribution to the increasing atmospheric methane (based on ethane/methane ratios) with the isotopic evidence. The title has been modified in accordance:

“Reduced biomass burning emissions reconcile conflicting estimates of the post-2006 atmospheric methane budget”

Comment (L14): I could understand this sentence only because I already knew the recent trends in methane and its isotopic signature. Needs rewording. Two sentences would be better.

Response: We revised the first sentence in the abstract to two sentences:

“Several viable but conflicting explanations have been proposed to explain the recent ~8ppb/yr increase in atmospheric methane after 2006, equivalent to a net emissions increase of ~25 TgCH₄/yr. A concurrent increase in atmospheric ethane implicates a fossil source; a concurrent decrease in the heavier isotopes of methane points towards a biogenic source, while other studies propose a decrease in the chemical sink (OH)”

Comment (L18): “Expected” is not appropriate in my opinion. Consider revising to “greater than twice the previously suggested ...”

Response: We would prefer to keep the word “expected” as we are referencing results from either the published literature or a standard community database (i.e. GFED). “Suggested” implies (at least to us) that these prior results were not based on a formal scientific inquiry.

Comment (L 28) These sentences are ambiguous. The importance of rising atmospheric methane levels, as it relates to the climate change problem, is primarily its direct radiative impact. That is not mentioned here. Things get more worrisome if the extra methane in the atmosphere is released from natural environments such as the permafrost or wetlands due to rising global temperatures. This also does not come through clearly. Should provide better context about the importance of methane and more specifically the importance of the recent rise.

Response: We have revised the opening sentences in the introduction (**line 30**) to address this comment. Also, we have broken up the first part of this sentence, removed some of the language, and added a statement on the role of methane as a GHG and ozone pre-cursor:

“Recent changes in the growth rate of methane¹, the second most important greenhouse gas and important ozone pre-cursor², could be due to changing anthropogenic emissions in the form of fossil fuel or agricultural emissions^{3,4,5,6,7,8}. Alternatively, natural wetland methane fluxes in the high-latitudes or tropics could be increasing in response to variations in temperature, the water

cycle and/or carbon availability to methanogens^{9,10,11,12} giving a preview of carbon cycle feedbacks to global warming¹³. However, determining the relative contributions of anthropogenic, biogeochemical, and chemical drivers of methane trends has been extremely challenging and consequently there”

Comment (L 44) Don't understand what is said here: sink changes can offset source changes or are you talking about different category of emissions? Isn't it always the case that there can be offsetting changes as long as the isotopes are satisfied.

Response: We removed this sentence as it was confusing and un-necessary.

Comment (L 113): It would be good to state the direction (heavier or lighter) of the recent update on isotopic signatures.

Response: Fixed

Comment (L 117): The isotope trajectories look exactly same to me although the uncertainties are different. Don't understand.

Response: This was due to a bug in the plotting code. We have fixed it and replaced the Figure.

Comment (L 123): Are the measurements shown in Fig. 3 an average of data from the sites shown in Supp-D (not S2)? This is not clear in Supp-D.

Response: We have updated our analysis based on the reviewer comments. A new supplemental section is added describing the updated method:

“Supplemental C: Calculation of Global average CH₄ and δ¹³C-CH₄:

Comment (L 135): Clarification needed. The way I see Figure 4, these increases are for going from GFED burning emissions to estimates presented in this study. It reads as if the numbers correspond to the effect of change in isotopic signatures.

Response: Figure 4 is intended to show how the fossil fuel and biogenic contributions to the increase change in response to different assumptions about the biomass burning emissions and for different assumptions about the isotopic signatures of the different sources. We have modified the figure caption accordingly:

“Figure 4: Change in average annual Biogenic and Fossil Fuel emissions between the 2001-2006 and 2007-2014 periods needed to fit the CH₄ mole fraction for different assumptions about biomass burning emissions and the isotopic signatures of the methane emission sources”

Comment (line 143) Are biofuel CH₄/CO emissions ratios different than biomass burning?

Response: For this study the emission factor is not relevant as we do not estimate methane emissions from biofuels using the estimated CO emissions. However, the isotopic values are the same as biomass burning.

Comment: (Line 231) Needs rewording and I don't understand the logic. Are you basically creating a regional emissions that would explain the CO observations (over oceanic sites?) given the chemistry/transport in the model. How does this mean treating each continent as a “region” and what does that even mean?

Response: We have attempted to clarify using updated text in Section M1.1 near **lines 262**. Basically, we assimilate MOPITT data over oceanic regions to effectively remove much of the

uncertainty due to errors in transport and chemistry from CO advected from the ocean to a continent. Much of our previous research has explored the role of transport and chemistry on CO and CH₄ fluxes (and also transport on CO₂ fluxes) and which have been discussed in many other papers on top-down flux estimates. The Jiang et al. 2013 and 2015 papers highlight many of the methods we have developed to address these errors. Here is the relevant text, hopefully this description makes sense!:

“We use a two-step approach to reduce the impact of these errors: firstly, we assimilate the MOPITT CO measurements over the ocean so that the modeled CO concentration fields that are advected over land from the ocean are consistent with the satellite data⁴². We then estimate the CO emissions through comparison of model and data just over continental regions. Effectively this approach accounts for advection of the observed CO fields over the continents from the oceans while reducing the sensitivity of emissions from one continent to those from other continents⁴².”

Comment (Line 237) What happens to possible disagreements between CO from different measurement techniques? Isn't that additional uncertainty?

Response: Different CO measurements (e.g. from different satellites) will have different measurement and systematic errors. For the work presented here we use three different CO estimates from the MOPITT data, with different vertical sensitivities and reported biases, to diagnose and minimize transport and chemistry errors on the CO flux estimates. The bias part of the systematic errors for MOPITT data is corrected by comparison of MOPITT data to aircraft data (see for example Deeter *et al.*, JGR 2013 for their validation of the MOPITT data). The disagreements between the CO fluxes, using the three different emissions estimate approaches as discussed in the text, are what we assume for the uncertainties. We have found from a large body of previous work (cited in the text), all using different CO measurements, that transport and chemistry error are the largest sources of error in these CO emission estimate studies. To provide further context to this issue, we have added additional citations to Jones *et al.*, (2003) and Kopacz et al. (2010) near **line 260** in the revised text as these studies use different satellite CO measurements but have to address the same problems related to transport and chemistry (note that the text stays mostly the same in this section).

Comment: (L 251) So, profile and lower-trop information provide very similar constraints. Do the results based on these two data sets agree?

Response: All three results broadly agree with each other and with previously published research (e.g. Yin *et al.*, 2015) using different approaches to regularize the CO emissions estimate. Where the disagreement is largest is in regions where we expect the largest uncertainties in transport (e.g. India and SE Asia). However, despite the disagreement, the Indian and SE Asian regions also show the same broad trends in fire emissions using the different inversion approaches. However, these regions do not contribute strongly to the global fire emissions budget which is primarily dominated by emissions from Africa, South America, and Indonesia. As noted in the previous Comment/Response above, we use the disagreement between these three CO emissions estimates as our uncertainty in the CO emissions.

Comment: (L272) Do you mean “errors” in the methane and CO emission factors are typically correlated? Otherwise, I don't understand what this means.

Response: We have removed this language as it was confusing.

Response to Reviewer 3

General Comment: This study addresses the key issue of the attribution of recent increase in CH₄ global concentrations to a specific process. Considering the very extensive amount of work behind this manuscript, the quality of the scientific approach and the conclusive elements brought by the authors, this manuscript must be considered for publication in Nature Communications.

The authors use complementary and robust approaches to quantify the contribution of biomass burning to the global emission trends and update contributions from fossil fuel and biogenic sources according to their new estimate of BB.

However, the current presentation and some missing details require major revisions. The following remarks and comments should be addressed before accepting the manuscript.

Response: We would like to thank the reviewer for their very encouraging review and comments. As suggested, we have re-organized the paper by bring in much of the supplemental section to the methods section. In addition, we have added discussion on the role of the GEOS-Chem model resolution on our conclusions.

Our responses follow:

General comments:

Comment 1. Structure

the efforts by the authors to fulfill Nature Communications' requirements in terms of format and length are acknowledged. However, the final shape of the submitted manuscript is an extremely dense and scattered text, hard to follow with the main text referring to the Methods and Supplement and conversely, with a logic sometime hard to follow by the reader.

All the scientific arguments, details on methods and materials, etc. are currently mostly included in the manuscript, but the authors are asked to reconsider the general structure of their manuscript. Please note that the text and method length limit is 5000 and 3000 words respectively, while the current manuscript has 2000 and 1800 words respectively, leaving space for a large part of the supplement of 2300 words.

In particular, Supplement B, C and D are key elements of the approach and should be included (at least partly with then reference to the Supplement, especially for the figures) in the Method section. The Supplement A.3 seems to present results and should then be included to the main body of the text.

Response: As discussed above, we have moved much of the supplement into the methods section. The paper by Jiang, Worden et al. (ACP 2017) is now published so we have removed Supplemental A and instead have added discussion of this paper to the methods section. The

discussion about the locations of the biomass burning reductions has been moved to the main text (near line 97) with the sentence: “*This decrease is largely accounted for by a 2.9 ± 1.2 TgCH₄/yr² decrease during 2006-2008, primarily due to a decrease in burning in Indonesia and South America*”.

Comment: Further details should be included in the main text to make it self-explaining and to bring additional details. For instance:

- please give more details on the numbers for the different contribution to the recent increase for the reader who do not know all the literature by heart; e.g., l. 76-77, the relative magnitude of the 1.5Tg/y becomes clear only after reading all the rest of the text

Response: We have removed the reference to the 1.5 Tg and the Schaefer et al. results here as it was confusing and instead reference GFED only and the approaches for quantifying CO emissions using bottom up approaches. Unfortunately, because these changes appear throughout the revised text, they are not copied to this comment/response section; we therefore urge the reviewer to re-read this section.

Comment:

- l. 116-117: please add details about the sink uncertainty:

Response: The discussion about the sink has been removed here and has been replaced by a more extensive discussion near **lines 167** in the revised paper as a more robust discussion of the role of the sink was also requested by the other reviewers.

Here is the revised paragraph:

*“Recent publications have also shown that we cannot rule out a decrease in the chemical sink of methane (reaction with OH) as the cause for the recent increase^{18,19}. To address this possibility, we have performed additional box model simulations where the sink is decreased progressively from 0% to 3%¹⁹ (Figure 5). The largest effect of assuming changes in the atmospheric OH sink is that the required global CH₄ source changes accordingly. For example, a 3% sink decrease would require a net source enhancement of 8 TgCH₄/yr instead of 25 TgCH₄/yr. The isotope source signature required to match the observed temporal evolution of δ^3C also changes, from -56‰ to -61‰. Using the mass balance equation (**Equation 10 Methods Section M2**), the corresponding FF emission contributions have been calculated for the different BB emissions change scenarios (iso-mf-OH scenario; Figure 5). We find that a FF enhancement of 3 - 9 TgCH₄/yr is still needed to explain the δ^3C measurements in case of a 3% OH sink decrease. Therefore, our conclusion that an increase in post-2007 FF emissions is needed to explain the observed shift in methane emissions⁵ remains valid, even if a sizeable fraction of the atmospheric methane concentration increase is due to decreasing atmospheric OH concentrations.”*

Comment: Impact of the model resolution on the results

The CO inversions are based on GEOS-Chem at 4x5 degrees. Although the inversion systems seems robust and smart adaptations are done to limit chemistry and transport errors, the model

resolution is quite different to the native resolution of MOPITT.
Are MOPITT measurements averaged over GEOS-Chem grid cells?

Response: Yes, as discussed in the Jiang et al. (2017 ACP) paper, the MOPITT data are averaged on the GEOS-Chem 4x5 (lat/lon) grid for every hour (to minimize potential transport representation errors). However, the emissions are calculated for monthly 4x5 estimates. We added this language, as well as language about the weighting for data and model in the inversion cost function, and a statement about averaging the emissions onto the GEOS-Chem grid cell in the first paragraph of Section M1.1.:

“The approach used to quantify CO fluxes over fifteen years using the GEOS-Chem adjoint and Terra MOPITT data is described in previously published research³¹. In summary, the inversion approach is to compare MOPITT data, averaged hourly and on the GEOS-Chem 5°x4° degree grid, to the model and modified by prior knowledge of CO emissions based on published inventories.”

Comment: Fire emissions are quite localized and generate thin plumes which are not represented with the 4x5 degrees resolution. Please discuss further the possible limitations due to the resolution.

Response: We have added discussion and citations (Jiang *et al.*, 2015, Stroud *et al.*, 2011) on the possible impacts of quantifying CO emissions at resolutions coarser than individual smoke plumes. In principal the CO emissions estimate of any grid cell will have an additional, not well characterized, error related to sub-grid scale diffusion and transport processes. However, as discussed by Jiang et al. (2015), the emissions from models at two different spatial resolutions become consistent when aggregated over large scales because they both have to be consistent with the observed CO concentrations. We add language to that effect near **lines 250**:

“As discussed in previous studies^{42,43}, observations or models that are coarser than the scales of the actual smoke plumes can have larger uncertainty because of the effects of sub-grid scale diffusion, transport processes, and chemistry. However, the emissions from models at different spatial resolutions that are observationally constrained by satellite concentration data become consistent when averaged over several of the coarser scale model grid cells because the different model posterior concentrations have to be consistent with the observed CO concentrations⁴². The emissions results presented here should therefore be conservatively interpreted as averages of all fire emissions over a month for aggregates of the GEOS-Chem grid cells (~2000 km spatial scales)”.

Comment Presentation

- The method section and Supplement should be of the same quality as the main body of the text. Some formulations are clumsy in this auxiliary sections.

Response: Hopefully combining supplemental with the methods fixed these issues. We have also checked the language of the supplemental.

- Please make notations in the equations consistent. In Sup. A, Eq. 2, vectors and matrices are not in bold font

Response: The correct notation has been implemented in the revised manuscript. Note that all values

Comment:- l. 306 of Methods, two times M3 title (**fixed**)

Comment: Table SD2: It would help the reader to present this table as Fig. 4

Response: Done

Comment: Sup. A1: prior emissions are deduced from different years and inventories, please comment the consistency of inventories.

Response: See response to reviewer 1. In summary, we re-normalize the GFED3 emissions used to calculate CO fluxes in Jiang et al. (2017) to GFED 4 using Equation 8 in Section M1.2

Comment: "data fusion approach": this doesn't sound like a standard name, from the reviewer's own knowledge

Response: removed language and referred to methods instead.

Comment: 69-70: the sentence is clumsy

Response: fixed

Comment: In M3, l. 321, where is tau used?

Response: We have updated the manuscript text to explain the role of tau.

Comment: l. 482-484, Figure 4: please reformulate to have a short description of the "no-change" experiment.

Response: We have updated the description of Figure 4 as per suggestion.

REVIEWERS' COMMENTS:

Reviewer #1 (Remarks to the Author):

Second review of

Reduced biomass burning emissions reconcile conflicting estimates of the post-2006 atmospheric methane budget

By John Worden et al.

This is the second round of reviews. The authors have adequately addressed all my previous concerns and suggestions.

A couple of very minor points should be cleared up before publication (see details below).

I note that there are some typos that I have not marked, assuming they will be taken care off in an editing step by Nature Comms.

Sincerely,

Hinrich Schaefer

Detailed comments:

Lines 179-181: (i) the fossil fuel enhancement is

(i) larger than the net source increase (line 175); the reader may benefit from a reminder that the total difference is net increase plus biomass burning reduction and

(ii) different from the value quoted in line 199. Please clarify.

Lines 162 and 197: the quoted values for biogenic enhancement differ slightly (rounding error?)

Lines 162-163; 180; 197; 199: it would probably be helpful to point out explicitly that these numbers are a split of the gross source enhancement between FF and biogenic emissions. Even then it is hard to immediately grasp the relative importance of either source type to the renewed increase. I recommend that the range of possible percentages for, e.g., FF emissions is given for the various scenarios.

Line 684: "... uncertainties in blue..."; not black?

Reviewer #2 (Remarks to the Author):

I find the author responses to my initial review generally satisfactory and do not see the manuscript again before publication. A couple of minor comments listed below.

Comment (L18): "Expected" is not appropriate in my opinion. Consider revising to "greater than twice the previously suggested"

Response: We would prefer to keep the word "expected"; as we are referencing results from either the published literature or a standard community database (i.e. GFED). "Suggested" implies (at least to us) that these prior results were not based on a formal scientific inquiry.

COMMENT: In my opinion, published results have to go through multiple rounds of formal scientific inquiry by independent researchers before the results become more than mere suggestions. Regardless, use of "expected" is fine as long as what it is expected from is explicitly stated and properly referenced in each instance of use.

Line 180: 3-9 TgCH₄/yr is the right range but looking at Fig. 5, I see 6-9 TgCH₄/yr.

Line 199: I think the 5-12 TgCH₄/yr FF emissions increase stated here should be the same as the number stated on line 180. Again looking at Fig. 5, if I include error bars, I see 3-12 TgCH₄/yr. Is there a mix up of numbers, may be?

Murat Aydin

Reviewer #3 (Remarks to the Author):

The authors made substantial efforts to address the comments of all reviewers. The manuscript is now clear to read and well organize.

It can be published as is.

Response to reviewers

This is the second round of reviews. The authors have adequately addressed all my previous concerns and suggestions.

A couple of very minor points should be cleared up before publication (see details below).

I note that there are some typos that I have not marked, assuming they will be taken care off in an editing step by Nature Comms.

Sincerely,

Hinrich Schaefer

Detailed comments:

Comment: Lines 179-181: (i) the fossil fuel enhancement is (i) larger than the net source increase (line 175); the reader may benefit from a reminder that the total difference is net increase plus biomass burning reduction and (ii) different from the value quoted in line 199. Please clarify.

Response: We have added ranges for the excess amount needed to explain increase, the sink, fires, and biogenic, and fossil fuels.

Comment: Lines 162 and 197: the quoted values for biogenic enhancement differ slightly (rounding error?)

Response (Fixed)

Comment: Lines 162-163; 180; 197; 199: it would probably be helpful to point out explicitly that these numbers are a split of the gross source enhancement between FF and biogenic emissions. Even then it is hard to immediately grasp the relative importance of either source type to the renewed increase. I recommend that the range of possible percentages for, e.g., FF emissions is given for the various scenarios.

Response: I understand the potential for confusion but it was not obvious if adding percentages would fix or only alter the confusion. Instead I opted for increased discussion of Figure 4.

Comment: Line 684: "... uncertainties in blue..."; not black?

Response (fixed)

Reviewer #2 (Remarks to the Author):

I find the author responses to my initial review generally satisfactory and do not see

the manuscript again before publication. A couple of minor comments listed below.

Comment (L18): Expected” is not appropriate in my opinion. Consider revising to "greater than twice the previously suggested"

Response: We would prefer to keep the word "expected"; as we are referencing results from either the published literature or a standard community database (i.e. GFED).

"Suggested" implies (at least to us) that these prior results were not based on a formal scientific inquiry.

COMMENT: In my opinion, published results have to go through multiple rounds of formal scientific inquiry by independent researchers before the results become more than mere suggestions. Regardless, use of "expected" is fine as long as what it is expected from is explicitly stated and properly referenced in each instance of use.

Response: We cite the GFED papers in the text. Hopefully that addresses this concern.

Comment: Line 180: 3-9 TgCH₄/yr is the right range but looking at Fig. 5, I see 6-9 TgCH₄/yr.

Response: Our apologies for the confusion. We are now taking the numbers from the top panel of Figure 5a (rightmost values) of between 6 -12 Tg CH₄/yr. (We could add another decimal place to these ranges but in the interest of clarification we have chosen to stick with integer ranges).

Comment: Line 199: I think the 5-12 TgCH₄/yr FF emissions increase stated here should be the same as the number stated on line 180. Again looking at Fig. 5, if I include error bars, I see 3-12 TgCH₄/yr. Is there a mix up of numbers, may be?

Response: Again, Our apologies for the confusion. We are now taking the numbers from the top panel of Figure 5a (rightmost values) of between 6 -12 Tg CH₄/yr.

Murat Aydin